# NanoVLA: Routing Decoupled Vision-Language Understanding for Nano-sized Generalist Robotic Policies

## Abstract

Vision-language-action (VLA) models have significantly advanced robotic manipulation by integrating vision-language models (VLMs), and action decoders into a unified architecture. However, their deployment on resource-constrained edge devices, such as mobile robots or embedded systems (*e.g.,* Jetson Orin Nano), remains challenging due to high computational demands, especially in real-world scenarios where power, latency, and computational resources are critical. To close this gap, we introduce **Nano**-scale **V**ision-**L**anguage **A**ction (NanoVLA), a family of lightweight VLA architectures that achieve high performance with minimal resources. Our core innovations include: (1) *vision-language decoupling* that moves conventional early vision and language inputs fusion in VLM to late stage, achieving better performance while enabling caching and reduce inference overhead and latency; (2) *long-short action chunking* to ensure smooth, coherent multi-step planning without sacrificing real-time responsiveness; (3) *dynamic routing* that adaptively assigns lightweight or heavy backbones based on task complexity, further optimizing inference efficiency. Experimental results on several benchmarks, as well as real-world deployments, demonstrate that NanoVLA achieves up to 52x faster inference on edge devices compared to previous state-of-the-art VLA models, with 98% less parameters while maintaining or surpassing their task accuracy and generalization. Ablation studies confirm that our decoupling strategy preserves cross-task transferability, and the routing module enhances cost-performance trade-offs, enabling practical, high-precision robotic manipulation on resource-constrained hardware.

## 1 Introduction

The conventional approach in robot learning has been to train task-specific models from scratch, but vision-language-action (VLA) models are emerging as a transformative paradigm (Brohan et al., 2022; Zitkovich et al., 2023; Mittal et al., 2023; Bjorck et al., 2025; Cheang et al., 2025). Built on pre-trained large language models (LLMs) (Yang et al., 2025a; Touvron et al., 2023; Team et al., 2024a; Cai et al., 2024) or vision-language models (VLMs) (Liu et al., 2023b; Chen et al., 2024; Beyer et al., 2024; Wang et al., 2024) with an appended action decoder, these models leverage web-scale multi-modal pretraining to ground natural language instructions in diverse scenarios, enabling rapid adaptation across tasks with fine-tuning. This paradigm brings broad generalization to robot learning, but it clashes with the realities of deploying on resource-constrained edge hardware (*e.g.,* Jetson Orin-class devices) for three key reasons: (1) inference remains slow and compute-intensive; (2) long-horizon behaviors often degrade into jerky or brittle motions; and (3) a single fixed backbone creates a mismatch between model capacity and task difficulty, resulting in overcomputing for simple actions and underperformance on long-horizon tasks. Consequently, the existing VLA models remain impractical outside datacenter-class machines (Wang et al., 2025).

In response, we introduce **Nano**-scale **V**ision-**L**anguage **A**ction (NanoVLA), a framework that closes this deployment gap by reorganizing where modalities are fused, how actions are unrolled over time, and when a larger model backbone is invoked. As shown in Figure 1, rather than simply shrinking model parameters, NanoVLA reframes the inference around three complementary ideas that together deliver low latency and robust behavior on edge devices:

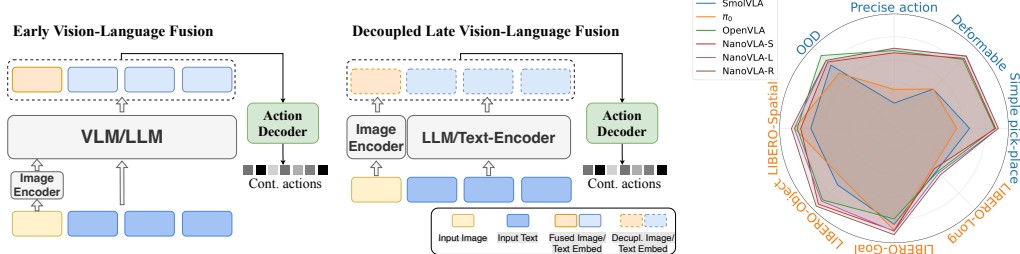

Figure 1: **Decoupled fusion for efficient VLA policies.** Our decoupled fusion strategy (mid) delays vision-to-language fusion in parameter-constrained settings. This approach does enables better performance with less overhead and latency, which informs NanoVLA, small scale VLA that achieves better performance across both simulation and real-world tasks with only ∼2% of the parameter of models like OpenVLA, as shown in the Radar plot (right).

- **Vision-language decoupling** (cache what does not change). Most VLAs repeatedly interleave vision and language with heavy cross-attention, forcing the language backbone to recompute at every control step, even when the instruction remains the same. NanoVLA keeps vision and language separate until the last stage, so instruction features can be encoded once and reused, while only the image embedding and action module are updated per frame, resulting less computation. Notably, pre-trained visual and language encoders have learned highly abstract and semantically rich representations independently; delayed fusion helps avoid early cross-modal interference while preserving the individuality of each modality (Shukor et al., 2025b). Experimental results show that this design maintains competitive performance while improving efficiency.

- **Long-short action chunking** (plan long but act short). Step-by-step predictors are reactive but often jittery; long fixed chunks are smooth but unresponsive (Bharadhwaj et al., 2024; Zhao et al., 2023). NanoVLA strikes a balance: the policy generates a longer chunk of actions, but *executes only a short window* before re-planning with fresh observations. This amortizes expensive planning over multiple control steps, while keeping behavior smooth and adaptable to new visual evidence.

- **Dynamic Routing** (small backbone by default, large backbone on demand). NanoVLA incorporates a lightweight router that directs simple tasks (e.g., short-horizon grasps) to a compact language backbone, escalating to a larger backbone only when task difficulty increases. This adaptive approach maintains low average latency while preserving performance on complex tasks.

Taken together, these three components follow a single principle: *spend compute only where it matters*. **Decoupling** avoids redundant cross-modal computation; **chunking** ensures smooth long-horizon behavior with reactivity, and **routing** matches model capacity to task difficulty. With these strategies, NanoVLA turns a generalist policy into a practical edge controller, efficient in both architecture and inference, rather than merely a downsized replica of a large model. In summary, to the best of our knowledge, the proposed NanoVLA is the first VLA framework to integrate decoupled, cache-friendly vision-language processing, chunked long-horizon control, and adaptive backbone routing into a single edge-oriented design. We list our main contributions as follows:

- We present a lightweight vision-language-action framework that makes large-model robot policies practical on edge devices by rethinking where modalities are fused, how actions are executed over time, and how compute is allocated.

- We demonstrate that our NanoVLA achieves a superior efficiency-effectiveness trade-off: it enables smooth long-horizon behaviors, low-latency inference, and adaptive capacity across varying task difficulties, capabilities not simultaneously supported by the existing VLAs.

- We validate NanoVLA on standard VLA benchmarks and real-world deployments, demonstrating that it matches or surpasses state-of-the-art performance while operating efficiently on resource-constrained hardware.

## 2 RELATED WORK

### 2.1 LARGE LANGUAGE MODELS FOR ROBOTICS

Large language models and their multi-modal extensions have recently become central to robot learning, which enables foundation robot policies that generalize across tasks, environments, and embodiments (Firoozi et al., 2025; Zhang et al., 2025b;a; Chen et al., 2025; Zhao et al., 2025; Zhou et al., 2025). By leveraging web-scale pretraining, such LLMs (Touvron et al., 2023; Beyer et al., 2024; Team et al., 2024a; Cai et al., 2024; Yang et al., 2025a) provide strong priors for semantic reasoning, while vision-language models (Liu et al., 2023b; Wang et al., 2024; Chen et al., 2024) leverage pre-trained visual encoder such as CLIP (Radford et al., 2021) and SigLIP (Zhai et al., 2023) to ground semantics in perception. This integration is most commonly realized in vision-language-action models, which pair visual encoders with LLM reasoning modules and action generators (*e.g.,* action tokenizer (Kim et al., 2024), DDPM diffusion model (Kim et al., 2025) and flow matching model (Gao et al., 2024)).

### 2.2 LARGE-SCALE GENERALIST POLICIES

The availability of massive robot datasets, most notably Open-X-Embodiment (Collaboration et al., 2023), has enabled the development of large-scale generalist policies that rely heavily on scaling both architectures and data. OpenVLA (Kim et al., 2024) exemplifies this trend with a 7B-parameter VLM model trained on nearly one million trajectories, achieving state-of-the-art generalization across a wide variety of robotic tasks. $\pi_0$ (Black et al., 2024) and $\pi_{0.5}$ (Black et al., 2025) leverage self-supervised pretraining on web-scale multi-modal, demonstrating better zero-shot and open-world generalization, highlighting the effectiveness of scaling for learning broad semantics and grounding actions across tasks. However, such large scale models makes them impractical for the deployment on resource-constrained platforms, where low latency and compute efficiency are important. This mismatch has driven growing interest in compact alternatives that preserve generalization while improving efficiency (Shukor et al., 2025a; Wen et al., 2025; Yang et al., 2025b).

### 2.3 COMPACT AND EFFICIENT VLAS.

More recently, efforts have centered on designing smaller VLA models that maintain generalization while reducing the computational cost of deployment. RT-1 (Brohan et al., 2022) pioneered the generalist robotic models with only 35M parameters. Octo-Base (Team et al., 2024b) introduces a lightweight, 90M-parameter transformer policy trained on 800k trajectories, serving as an efficiency-oriented baseline. SmolVLA (Shukor et al., 2025a) takes this further by distilling knowledge from large VLAs into a compact 500M-parameter model, demonstrating that downsizing can preserve much of the generalization ability while reducing inference costs. HiRT Zhang et al. (2024) is a hierarchical fast-slow design: a large VLM (InstructBLIP, 7B) runs at a low frequency to produce multimodal latents, and a small, high-frequency visual policy executes actions asynchronously, guided by those slowly updated features. The VLM's outputs are cached in a latent buffer and refreshed periodically. Practically, HiRT's improvements still depend on periodic 7B VLM inference which does not allow it to be execute on resource constrained devices. These works signal a growing recognition that practical VLA deployment must address computational efficiency and edge hardware constraints, not only performance on datacenter GPUs.

## 3 NANOVLA

Deploying VLA models on edge devices requires more than shrinking parameters. It demands rethinking inference. The existing VLAs suffer from redundant cross-modal processing, brittle open-loop action execution, and fixed backbones that cannot adapt to task difficulty. To address these issues, **NanoVLA** introduces three design choices: (i) **vision-language decoupling with caching**, which avoids recomputing static instructions while updating visual inputs; (ii) **long-short action chunking**, which plans over long horizons but executes with frequent feedback; and (iii) **dynamic routing**, which allocates computation adaptively by switching between lightweight and heavyweight backbones based on task complexity.

## 3.1 VISION-LANGUAGE DECOUPLING WITH CACHING

Modern VLA models demand heavy computation because they tightly interleave modalities through repeated cross-attention, forcing both vision and language backbones to recompute at every timestep. This design hinders real-time deployment on edge devices, where instructions often remain fixed while only the visual observations change. The existing work has shown that freezing large pre-trained encoders and using lightweight adapters can preserve accuracy while reducing computation (Li et al., 2023). Building on this insight, our NanoVLA introduces a decoupled architecture that processes vision and language independently until a late fusion stage.

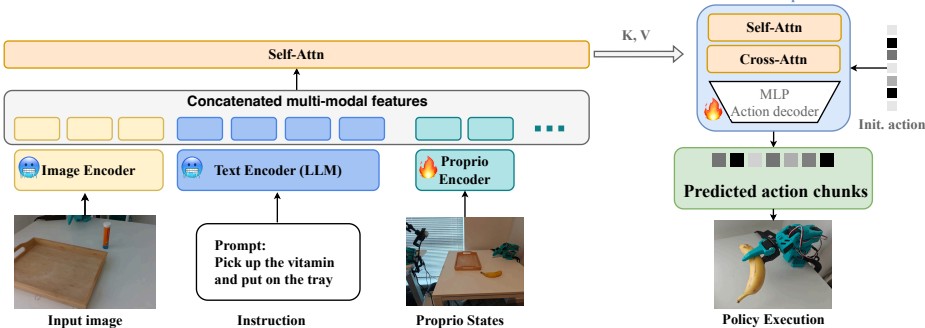

Figure 2: Overview of NanoVLA framework. Multi-modal inputs are processed independently and fused at a late stage via a lightweight attention layer. This design bypasses the compute-intensive early fusion in VLMs, unlocking key advantages like caching and accelerated inference.

**Method.** NanoVLA separates each input modality into its own encoder: a visual encoder (*e.g.,* ResNet (He et al., 2016), ViT (Dosovitskiy et al., 2020)) extracts compact scene features, while a language encoder (e.g., BERT (Devlin et al., 2019), Qwen (Yang et al., 2025a)) encodes task instructions. Both encoders remain frozen during training, preserving their pretrained semantic knowledge and avoiding catastrophic forgetting. Additional modalities, such as proprioception, are incorporated via lightweight MLP projections and fused with self-attention. At the action-generation stage, we merge the modality-specific embeddings using a lightweight transformer. Each layer first applies self-attention over the output tokens, allowing the model to capture dependencies within the multi-modality features. Then, it applies cross-attention that fuses the visual and language features for action generation (see Appendix for details A.2). Because cross-modal attention is performed only once at this late stage, this design dramatically reduces redundant computation. This late fusion design bypasses the compute-intensive early cross-modal entanglement common in VLMs.

**Caching for efficiency.** A key advantage of this separation is the ability to cache intermediate features. For example, in an interactive robot setting, the instruction embedding needs to be computed only once and reused across timesteps, while the visual embedding is updated at every frame. This caching eliminates redundant computation and significantly reduces on-device latency, making inference practical on resource-constrained hardware.

The pretrained visual and language encoders produce rich and high-level semantic representations, as also observed in FrEVL (Bourigault & Bourigault, 2025), which the lightweight decoder can effectively fuse for accurate action prediction. By aligning computation with modality dynamics, which reuses stable instruction features while refreshing changing visual features, NanoVLA achieves competitive accuracy with far fewer trainable parameters and much faster inference.

## 3.2 LONG-SHORT ACTION CHUNKING

A second bottleneck for edge deployment arises from how actions are generated over time. Conventional chunking strategies (Zhao et al., 2023; Kim et al., 2025) predict a sequence of actions and then execute the entire chunk in an open-loop. While this reduces the number of forward passes, it introduces a critical weakness: once a chunk is committed, the robot cannot correct for model mismatch, sensing latency, or environmental changes until the next replan. As a result, behaviors often become jerky, unstable, or misaligned in long-horizon tasks. Typically, real-world control requires

two conflicting properties: *Temporal coherence*: Actions unfold smoothly across long horizons; *Responsiveness*: The policy can adapt quickly to new sensory input.

A fixed chunk size cannot satisfy both. If chunks are long, execution is smooth but brittle; if chunks are short, execution is reactive but jittery. Our goal is to retain the benefits of long-horizon planning while preserving frequent feedback during execution.

**Method.** In response, we propose **long-short action chunking (LSAC)**. During training, the policy is optimized to predict long sequences of actions, capturing temporal patterns and dependencies across horizons:

$$\mathbf{a}_{t:t+H_{\text{train}}-1} = \pi_\theta(x_t, l), \tag{1}$$

where $x_t$ is the current observation, $l$ is the task instruction, and $H_{\text{train}}$ is the planning horizon. The training objective is supervised regression across the full chunk:

$$\mathcal{L}_{\text{chunk}}(\theta) = \frac{1}{H_{\text{train}}} \sum_{k=0}^{H_{\text{train}}-1} \ell(\pi_\theta(x_t, l)_k, \ a_{t+k}), \tag{2}$$

where $\ell$ is an $\ell_1$ or $\ell_2$ loss. At inference time, however, the robot executes only the first $h \ll H_{\text{train}}$ actions of each predicted chunk before replanning with the latest observation:

$$\hat{\mathbf{a}}_{t:t+h-1} \leftarrow \pi_\theta(x_t, l)_{0:h}, \qquad t \leftarrow t + h, \text{ repeat}. \tag{3}$$

This develops a long-short mismatch: the model plans over long horizons but acts in short bursts with frequent replanning. Overall, LSAC balances coherence and reactivity in long-horizon control. Training on long sequences promotes **smoothness**, executing short segments with replanning ensures **adaptability**, and predicting many steps per pass while acting on a few delivers **efficiency**. These properties together make LSAC well-suited for real-world robotic control on edge devices.

### 3.3 DYNAMIC ROUTING

A final challenge for efficient VLA deployment on edge devices is the mismatch between task difficulty and model capacity. A single fixed backbone wastes computation on simple instructions (*e.g.*, short-horizon grasps) yet may still struggle with complex and reasoning-heavy tasks. What is needed is a mechanism to adaptively allocate compute, using lightweight models when possible and escalating to heavier ones only when necessary.

**Method.** Our NanoVLA introduces a **VLA router** that selects among candidate language models at inference time. Instead of relying on hard labels for routing (Ong et al., 2024), we estimate uncertainty-aware win probabilities between models. For each task $l \in \mathcal{T}$ and model $m \in \mathcal{M}$, we observe $n_{m,l}$ trials with $s_{m,l}$ successes, producing an empirical success rate $\hat{p}_{m,l} = \frac{s_{m,l}}{n_{m,l}}$. A router $R$ chose a model for a new task $l$, trading off predicted performance and cost.

**Bayesian success modeling.** Rather than treating $\hat{p}_{m,l}$ as a point estimate, we place a Beta-Binomial model on the (unknown) per-task success probability to model uncertainty:

$$p_{m,l} \sim \text{Beta}(\alpha_{m,l}, \beta_{m,l}), \qquad \alpha_{m,l} = s_{m,l} + \alpha_0, \beta_{m,l} = (n_{m,l} - s_{m,l}) + \beta_0, \tag{4}$$

with a weakly informative prior (we use $\alpha_0 = \beta_0 = 1$ unless stated otherwise). This posterior reflects trial-count uncertainty: with few trials, posteriors are wider and induce probabilities closer to $1/2$ in the pairwise comparisons below. Let $\mathcal{D}$ denote all observed trials.

**Pairwise win probability.** For any two models $i, j \in \mathcal{M}$ on task $l$, the win probability is defined:

$$\pi_{i \succ j}(l) = \Pr\left(p_{i,l} > p_{j,l} \big| \mathcal{D}\right) = \int_0^1 \int_0^1 \mathbb{I}_{p_{i,l} > p_{j,l}} \text{Beta}\left(p_{i,l}; \alpha_{i,l}, \beta_{i,l}\right) \text{Beta}\left(p_{j,l}; \alpha_{j,l}, \beta_{j,l}\right) dp_{i,l} dp_{j,l}. \tag{5}$$

When $\alpha_{i,l} \in \mathbb{N}$, $\pi_{i \succ j}(l)$ admits the finite-sum closed form

$$\pi_{i \succ j}(l) = \sum_{k=0}^{\alpha_{i,l}-1} \binom{\beta_{i,l} + k - 1}{k} \frac{B\left(\alpha_{j,l} + k, \beta_{i,l} + \beta_{j,l}\right)}{B\left(\alpha_{j,l}, \beta_{j,l}\right)}, \tag{6}$$

where $B(\cdot, \cdot)$ is the Beta function. To support non-integer hyperparameters uniformly, our **Monte Carlo-Beta (MCB)** estimator uses Monte Carlo: draw $x^{(r)} \sim \text{Beta}(\alpha_{i,l}, \beta_{i,l})$ and $y^{(r)} \sim \text{Beta}(\alpha_{j,l}, \beta_{j,l})$ for $r = 1, \ldots, R$ and set

$$\widehat{\pi}_{i \succ j}(l) = \frac{1}{R} \sum_{r=1}^{R} \mathbb{I}_{x^{(r)} > y^{(r)}}. \tag{7}$$

We use the closed form equation 6 when applicable and otherwise fall back to the Monte Carlo estimator equation 7. Targets are clipped to $[\varepsilon, 1 - \varepsilon]$ with $\varepsilon = 10^{-4}$ for numerical stability.

**Training objective.** We train a text-conditioned binary classifier $f_\theta$ on serialized inputs $(l, i, j)$ to predict $\Pr(i \succ j \mid l)$. Given the MCB target $\widehat{\pi}_{i \succ j}(l)$, we minimize the Bernoulli log-loss

$$\mathcal{L}\text{pair} = -\mathbb{E}_l, \mathbb{E}_{i<j} \left[ \widehat{\pi}_{i \succ j}(l) \log \sigma\left(z_{i \succ j}(l)\right) + \left(1 - \widehat{\pi}_{i \succ j}(l)\right) \log \left(1 - \sigma\left(z_{i \succ j}(l)\right)\right) \right], \tag{8}$$

where $z_{i \succ j}(l)$ is the model logit and $\sigma$ is the logistic function. The win probabilities provide soft, calibrated supervision targets for training a text-conditioned binary classifier $f_\theta$ to predict $\Pr(i \succ j \mid l)$. At inference time, the router defaults to the lightweight model. It only escalates to the large model if the predicted success probability of the large model for task $l$, $\hat{p}_L(l)$, exceeds a threshold $\tau$, which improves efficiency on simple tasks, and preserves accuracy on complex instructions.

## 4 EXPERIMENTS

To comprehensively evaluate our model's performance, we conducted experiments across a wide range of environmental setups, including 5 tasks in simulation environment and 12 tasks on real robots. The implementation details are provided in Appendix A.9, where we have three versions of NanoVLA, *NanoVLA-S* uses BERT-base as the text backbone, *NanoVLA-L* uses Qwen2.5 0.5B as the text backbone, and *NanoVLA-R* is the Router version. All models were using ResNet18 as the image encoder, and were trained with 100 AC steps with 10 AC steps during inference. We benchmark our approach against the following generalist policies: *Octo-Base* (Team et al., 2024b), *OpenVLA* (Kim et al., 2024), $\pi_0$ (Black et al., 2024), *TraceVLA* (Zheng et al., 2024), *SpatialVLA* (Qu et al., 2025), *SmolVLA* (Shukor et al., 2025a).

### 4.1 SIMULATED BENCHMARK

**LIBERO** is a standardized lifelong learning benchmark for VLA models (Liu et al., 2023a). In this experiment, we follow OpenVLA to use tasks from *LIBERO-Spatial*, *LIBERO-Object*, *LIBERO-Goal*, and the *LIBERO-Long*. These suites respectively emphasize spatial relations, object variation, goal (end-state) variation, and mixed/entangled tasks. We report the success rate (SR) of NanoVLA models as the average of 50 independent trails. For OpenVLA, TraceVLA, SpatialVLA, and Octo, we report the SR from their original paper. For $\pi_0$ and SmolVLA, we replicate the experiment from LeRobot code base with default setting (Cadene et al., 2024) and report the average SR, due to $\pi_0$ original paper did not provide the LIBERO results.

**Overall Performance.** As shown in Table 1, NanoVLA consistently outperforms multi-Billion Parameter VLA models, such as OpenVLA, $\pi_0$, TraceVLA, and SpatialVLA in the LIBERO tasks. NanoVLA shows significant performance gains, with improvements ranging from 6.0% to 12.3% in average success rates while only using less than 10% total parameters. When compared to other baselines like Octo-Base and SmolVLA, all NanoVLA models generally perform better, with fewer trainable parameters needed. These results suggest that the proposed NanoVLA framework can achieve generalized policy execution across various robotic manipulation tasks with significantly reduced model parameters, leading to improved performance-throughput trade-off in edge side executions. Especially, the proposed NanoVLA-R can further improve the precision by 3.7% while reducing 43% of the parameters of NanoVLA-L.

As for For OpenVLA-OFT and UniVLA, we view these models as upper bounds rather than direct competitors: they use 7-8.5B total parameters (more than 25–30× larger than NanoVLA-R's 296M) and still require more trainable parameters (100M or all parameters) than our 52M trainable parameters. As a result, it is unsurprising that OpenVLA-OFT and UniVLA achieve the best absolute

| Model type | Policy | Total Params | Trainable Params | Spatial | Object | Goal | Long | Avg. |
|---|---|---|---|---|---|---|---|---|
| Data center scale VLAs | OpenVLA | 7.5B | 279M | 84.7 | 88.4 | 79.2 | 53.7 | 76.5 |
| | $\pi_0$ | 3.5B | 3.1B | 96.8 | 98.8 | 95.8 | 94.2 | 94.2 |
| | TraceVLA | 7B | 7B | 84.6 | 85.2 | 75.1 | 54.1 | 74.8 |
| | SpatialVLA | 3.5B | 50M | 88.2 | 89.9 | 78.6 | 55.5 | 78.1 |
| | OpenVLA-OFT | 7B | 100M | 96.2 | 98.3 | 96.2 | 90.7 | 95.3 |
| | UniVLA | 8.5B | - | 95.4 | 98.8 | 93.6 | 94.0 | 95.5 |
| Efficient VLAs | Octo | 90M | 90M | 78.9 | 85.7 | 84.6 | 51.1 | 75.1 |
| | SmolVLA | 450M | 100M | 72.8 | 69.8 | 84.0 | 52.6 | 78.6 |
| | NanoVLA-S | 161M | 52M | 81.6 | 93.6 | 89.6 | 49.8 | 78.7 |
| | NanoVLA-L | 520M | 52M | 87.2 | 89.8 | 90.0 | 55.2 | 80.4 |
| | NanoVLA-R | 296M* | 52M | **89.8** | **96.2** | **93.0** | **57.4** | **84.1** |

Table 1: Success rates (%) on four LIBERO task suites. Overall performance is calculated as the average over 50 test trials. (*total parameters calculated based on average L/S model invocation.)

performance (95.3% and 95.5% Avg.), but importantly, NanoVLA-R remains within 11% SR of these models despite its much smaller capacity. This shows that NanoVLA closes most of the gap to the largest VLAs while operating in a radically different efficiency regime, which is crucial for deployment on resource-constrained robots rather than data-center hardware.

**LIBERO-90.** We further evaluate on *LIBERO-90* (90 short-horizon tasks), the official split of *LIBERO-100* designed to evaluate scalability and generalization across a broader and more diverse task distribution than the traditional four-suite setting. As shown in Table 2, our NanoVLA-L achieves the highest success (83.3%), with large absolute gains over compact and widely reported baselines (e.g., +14.4% over SmolVLA at 68.9% and +21.3% over OpenVLA at 62.0%), indicating stronger instruction following across many short-horizon skills rather than overfitting to a small set of controlled shifts. Notably, NanoVLA-S, which uses a weaker language backbone, performs worst among our variants (55.1%), suggesting the limitations of encoder-only language models in handling large-scale instruction following tasks. NanoVLA-R, also achieved 81.6% SR, which sacrificed 1.7% performance compared to NanoVLA-L while reducing the average total parameters by 41.0% to 307M.

| **Model** | SmolVLA | OpenVLA | NanoVLA-S | NanoVLA-L | NanoVLA-R |
|---|---|---|---|---|---|
| LIBERO-90 | 68.9 | 62.0 | 55.1 | **83.3** | 81.6 |

Table 2: Results on LIBERO-90 benchmark (%).

## 4.2 REAL-WORLD VALIDATION

We evaluate NanoVLA on the easy to access and low cost LeRobot and edge-side device Jetson Orin Nano (8GB). Below we introduce the experiment set up for the robots and the results.

**LeRobot.** As shown in Figure 3, we assembled LeRobot So-arm 101 for robot manipulation tasks using a fixed-mounted third-person view camera capturing default $1280 \times 720$ RGB images. We collected 50 demonstration trajectories per task and merged them together for finetuning. The detailed real-world experimental setup and protocols are presented in Appendix A.3. All real-robot models were finetuned on the same data mixture, 50 demonstrations per task across our 10 tasks, using the same action horizon/chunking protocol for fairness.

As shown in Table 3, NanoVLA consistently outperforms the baseline across diverse tasks including pick-and-place operations, precise object movement, and deformable object manipulation. Notably, in easier pick-place tasks, NanoVLA-S (161M parameters) can even outperform OpenVLA (7B parameters), which has 43.5x more parameters. For deformable object such as banana and towel, all NanoVLA-based models achieved 90%+ success rate, beating all baseline. Additionally, for more complex tasks such open/close lid that require precise action generation, Nano-VLA can also consistently outperform baselines. Furthermore, we evaluate the NanoVLA's generalization capa-

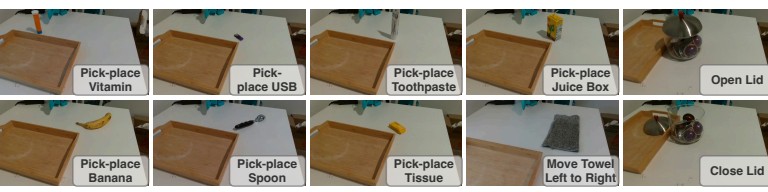

Figure 3: **LeRobot experiment setup**. We design 10 real-world tasks with different manipulation skills and objects with additional 2 unseen tasks for evaluating policy generalization.

bilities, we conducted additional experiments with two out-of-domain (OOD) tasks for both unseen tasks and unseen objects. The results in Table 3 demonstrates NanoVLA generalization capability on unseen tasks due to its superior capability in language grounding.

Additionaly, the OOD pick-place task show that the Router (NanoVLA-R) achieves 84.0% success, essentially identical to always using NanoVLA-L (84.0%) and clearly better than the small expert NanoVLA-S (82.0%), showing that the router does not degrade accuracy even when some tasks are out-of-distribution. This suggests that, although the win probabilities may be less well calibrated on OOD tasks, they are still accurate enough in a relative sense: whenever the router is unsure how an OOD task relates to the training distribution, it defaults to the safer large expert, so NanoVLA-R matches NanoVLA-L's performance while still routing many in-distribution tasks to cheaper experts.

| | Task | SmolVLA | $\pi_0$ | OpenVLA | Nano-S | Nano-L | Nano-R |
|---|---|---|---|---|---|---|---|
| Simple pick-place | Pick-place vitamin | 64.0 | 36.0 | 90.0 | **96.0** | **96.0** | **96.0** |
| | Pick-place spoon | 58.0 | 46.0 | 88.0 | **92.0** | 90.0 | **92.0** |
| | Pick-place USB drive | 80.0 | 64.0 | 74.0 | 76.0 | **82.0** | 80.0 |
| | Pick-place toothpaste | 84.0 | 76.0 | **94.0** | 88.0 | 92.0 | 92.0 |
| | Pick-place juice box | 44.0 | 52.0 | 94.0 | 92.0 | **96.0** | **96.0** |
| Deformable | Pick-place banana | 32.0 | 40.0 | 86.0 | **94.0** | 90.0 | 92.0 |
| | Pick-place tissue | 78.0 | 62.0 | 90.0 | 92.0 | **98.0** | **98.0** |
| | Move towel right | 22.0 | 34.0 | 68.0 | 66.0 | **70.0** | **70.0** |
| Precise | Open lid | 54.0 | 48.0 | 56.0 | 54.0 | **76.0** | 72.0 |
| | Close lid | 44.0 | 42.0 | 54.0 | 42.0 | 68.0 | **70.0** |
| ood | OOD pick-place | 78.0 | 68.0 | **90.0** | 82.0 | 84.0 | 84.0 |
| long | Long horizon task | 62.0 | 70.0 | **85.0** | 76.0 | 78.0 | 78.0 |
| | **Avg.** | 58.3 | 53.2 | 80.8 | 79.2 | **85.0** | **85.0** |

Table 3: Performance comparison of NanoVLA (*i.e.,* Nano-S, Nano-L and Nano-R) models and baselines on 12 real-world LeRobot manipulation tasks.

We additionally evaluate NanoVLA on a long-horizon manipulation task in a heavily cluttered tabletop environment. In this scenario, the robot must reason over multiple sequential sub-tasks approaching the target, maneuvering around distractors, grasping, and placing, while avoiding collisions with clutter. Figure 4 visualizes one successful policy execution: the yellow–orange–red trajectory shows the predicted end-effector path, and the frames are sampled at key manipulation events along the episode. This qualitative example demonstrates that NanoVLA can maintain coherent behavior over many steps and operate reliably in visually complex, cluttered scenes, going beyond simple single-object pick-and-place tasks.

## 4.3 INFERENCE LATENCY ANALYSIS

To validate the effectiveness of the edge-side deployment for NanoVLA family, we deployed them on the *Jetson Orin Nano Super Developer Kit*, which has 8GB memory with CUDA support and a computation performance for up to 67 TOPS. For fair comparison, we report the results in following sections on publicly available LIBERO-Goal. As shown in Figure 5a, NanoVLA outperforms Open-VLA, achieving 52x higher FPS and +13.8% SR. Please note that even the 4-bit quantized Open-

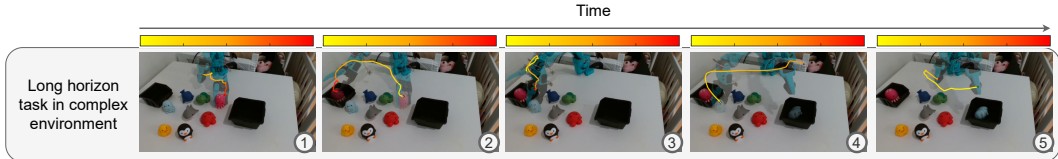

Figure 4: Long-horizon manipulation task in a complex environment. The yellow lines indicates the predicted end-effector trajectory over time, and the snapshots are taken at key manipulation events.

VLA model runs out of memory on our hardware, we could not measure its end-to-end throughput directly. Instead, we approximate its runtime by benchmarking the 4-bit LLaMA2 model that serves as OpenVLA's text backbone. We measure the wall-clock time of the 4-bit LLaMA2 forward pass on our evaluation hardware and compute FPS as the number of processed frames divided by this time. This measurement does not include the image encoder, and we only time the language backbone. In practice, the full OpenVLA pipeline must run both the image encoder and the LLaMA2 backbone, so its actual FPS would be lower than our estimate. Therefore, the reported 52× speedup of NanoVLA over OpenVLA should be interpreted as a conservative lower bound on the true end-to-end speedup.

When comparing the smaller SmolVLA, NanoVLA with 10 action-chunk (AC) steps is only slightly slower (-17.1%) than SmolVLA at 50 AC steps; when matching 50 AC steps, NanoVLA shows a minor SR decrease from ∼90% to 87.2%, which is still 3.2% above SmolVLA, while delivering 43.8% higher FPS. We also evaluate the training efficiency of NanoVLA, as detailed in Appendix A.6.

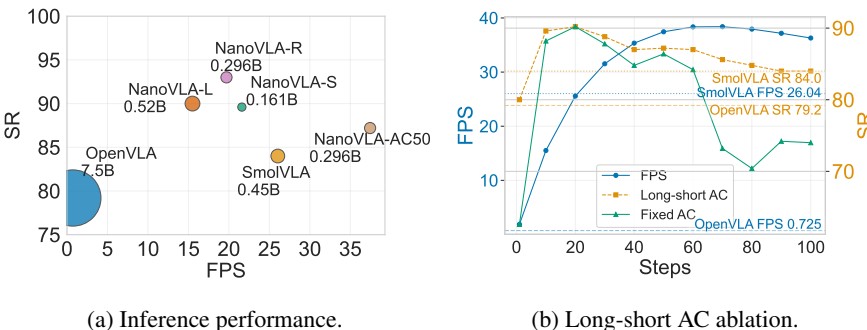

(a) Inference performance.

(b) Long-short AC ablation.

Figure 5: Inference analysis on Jetson Orin Nano (left) and long-short action chunking ablations.

## 4.4 ABLATION STUDIES

**Long-short action chunking effectiveness.** We also compare success rate for traditional fixed action-chunking (AC) and our long-short AC while also reporting throughput (FPS) across step sizes. As shown in Figure 5b, both methods peak at 20 steps (90.2 SR), but their behaviors diverge thereafter. Long-short AC maintains a high, flat SR plateau from 20-60 steps, then decays gently to 84.0 by 100 steps. Fixed AC is unstable beyond 30-40 steps, dropping sharply after 60 to 70-74 SR. Additionally, throughput increases with step size for both methods, which yields favorable operating points where long-short AC simultaneously surpasses the SmolVLA reference. Overall, long-short AC offers a superior SR-throughput trade-off and markedly better stability across step sizes. It's also more flexible in real-world deployment as it only need to train once. It preserves near-peak SR over a wide range while allowing the system to dial up steps to increase FPS with only modest SR loss, whereas fixed AC suffers a steep accuracy collapse at larger steps.

**Environmental Variant Impact.** Figure 6a demonstrates environmental conditions' impact on policy execution. We collected the real-world training data during night time, and conducted the test in both day and night time. Apart from lighting condition changes, robot arm and goal object's shadow will impact the image understanding. Notably, NanoVLA shows substantial enhancements over baselines, with an SR improvement between 8.5% to 42.1% in average performance. The use of continuous action decoder while freezing decoupled LLM and image encoder proves particularly effective when encountering the environmental shift, as the LLM generated action embeddings won't

be significantly impacted by the image embeddings, which can provide robust trajectory generation. This helps the model maintain policy execution SR despite changes in environments.

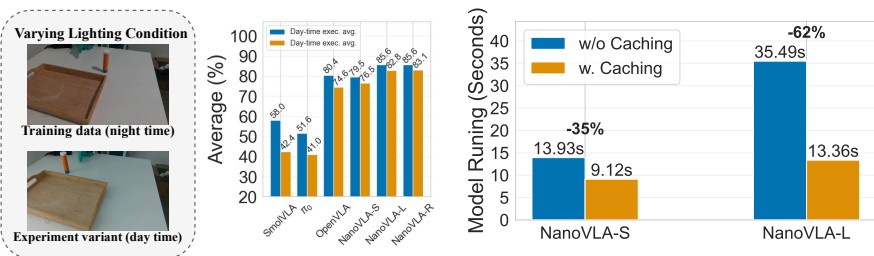

(a) Environmental variant impact

(b) LLM caching effectiveness

Figure 6: Ablation studies on environmental variant impact (left) and caching effectiveness (right).

**LLM caching throughput.** As shown in Figure 6b, when using Qwen 0.5B as the backbone, caching mechanism can save 62% inference time per execution task due to the LLM decoupling mechanism in the NanoVLA. Additionally, for smaller backbone such as BERT-base, we also observed a 35% inference time reduction, suggesting the efficiency of the decoupling method.

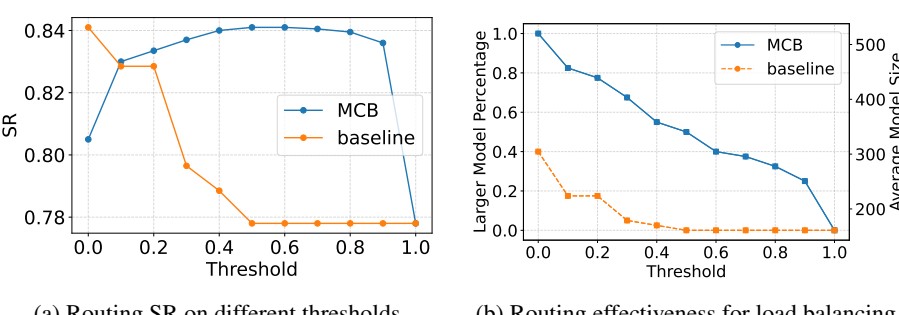

(a) Routing SR on different thresholds

(b) Routing effectiveness for load balancing.

Figure 7: MCB Routing Effectiveness

**Routing efficiency.** We analyze the effect of the decision threshold on SR-compute trade-offs for the proposed MCB router. Figure 7a presents Routing SR on LIBERO task suites under different thresholds. As a result of routing, the SR increases from 0.805 (NanoVLA-L only) at $\tau$=0 to a broad plateau around 0.84 spanning $\tau \in [0.4, 0.9]$ (mixture of NanoVLA-S/L. Additionally, as presented in Figure 7b, the routing module successfully reduces the average model size from NanoVLA-L's 520M to 251M while increasing the performance from 80.5% to 83.6%. it also significant increases NanoVLA-S's performance by 5.8% from 77.8% to 83.6%, while only increases the average number of parameters from 161M to 251M. This presents effectiveness of the proposed MCB router in selecting most effective model in various tasks. When comparing to the naive baseline which selects the model purely based on SR, the proposed MCB approach present a wider operating range, while the baseline is highly sensitive to threshold choice. MCB is robust to mis-calibration and task heterogeneity obtain near-maximal precision based on the computational resources, whereas the baseline requires staying very close to $\tau = 0$ to avoid a substantial accuracy penalty.

## 5 CONCLUSION

We presented NanoVLA, a lightweight VLA framework that makes foundational robot policies practical on edge devices. Through decoupling with caching, long-short chunking, and dynamic routing, it delivers smooth long-horizon behaviors, low-latency inference, and adaptive compute allocation. Experiments on benchmarks and real robots demonstrate that NanoVLA matches or surpasses state-of-the-art VLAs while running efficiently on resource-constrained hardware, offering a practical path toward scalable real-time embodied AI.

# 6 ETHICS STATEMENT.

All authors have read and agree to abide by the ICLR Code of Ethics. Our study focuses on robot manipulation in simulation (LIBERO) and in a controlled lab environment. No human subjects data were collected; the real-robot scenes contain only the robot, workspace, and household objects. No personally identifiable information is present, and no biometric data are processed. Real-robot experiments were conducted under institutional safety procedures with a physical emergency stop, speed/torque limits, collision monitoring, and a clear human-robot separation zone. We release code and models for research and educational use. Datasets used (e.g., LIBERO task suites) are publicly available under their respective licenses; we respect third-party licenses and trademarks that may appear incidentally on consumer products in the scenes. While our tasks do not involve sensitive attributes, dataset composition may still induce distributional biases (object categories, textures, lighting). We discuss failure modes and robustness (e.g., to lighting, occlusions, and slipped grasps) and provide evaluation details to support responsible interpretation of results. We estimate that our approach has a comparatively small environmental footprint due to its lightweight architecture; we report hardware, training duration, and energy considerations in the appendix to aid assessment. The authors declare no conflicts of interest.

# 7 REPRODUCIBILITY STATEMENT.

We make a concerted effort to enable full reproducibility. The paper specifies the model and training procedure, with complete hyperparameters summarized in the appendix (see "Hyperparameters" table) and the exact configuration files included as supplementary material. We provide an anonymized repository containing training and evaluation scripts, pinned dependencies (environment files), and instructions for dataset acquisition and preprocessing (including LIBERO splits). Where applicable, we include ablation settings, exact prompts/instructions, and checkpoints used to produce figures and tables. A step-by-step reproduction guide (shell commands) is provided in the appendix for edge device deployment; additional artifacts (logs, checkpoints) are referenced in the supplementary materials.

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

# A APPENDIX

## A.1 LLM USAGE.

In the interest of transparency, we disclose the use of LLM as a general-purpose assistive tool for writing. The LLM was used for grammar polish and tables formatting. It was not used for research ideation, experimental design, algorithm development, data collection/labeling, result selection, or quantitative analysis. All technical content, methodology, experiments, and conclusions are the authors' own; all claims and numbers were verified by the authors.

## A.2 TRANSFORMER ARCHITECTURE AND LATE-FUSION DETAILS

**Notation.** Let $D$ be the latent dimension, $h$ the number of heads, $d_k = D/h$. The encoder operates over a token sequence $\mathcal{X} = [x^{\text{lat}}, x^{\text{state}}, x^{\text{env}}, x^{\text{lang}}, X^{\text{img}}]$ where the first four are single 1D tokens and $X^{\text{img}}$ is a grid of image tokens flattened to length $N_{\text{img}}$. We denote the encoder output by $H \in \mathbb{R}^{N \times D}$ and the decoder runs on $H$ action query slots $Y \in \mathbb{R}^{H \times D}$ ("chunk size" in code).

**Input Projections and Token Layout** Each 1D modality is projected to $D$ by a linear layer:

$$x^{\text{state}} = W_{\text{state}}\, s, \quad x^{\text{env}} = W_{\text{env}}\, e, \quad x^{\text{lang}} = l, \quad x^{\text{lat}} = W_{\text{lat}}\, z,$$

with $z$ initialized as zeros at inference and sampled from a VAE posterior during training. Visual features come from a frozen backbone (e.g., ResNet) and are $C \times H \times W$ maps projected by a $1 \times 1$ conv $W_{\text{img}}$ to $D$ per spatial location, then flattened to tokens $X^{\text{img}} \in \mathbb{R}^{N_{\text{img}} \times D}$.

**Positional Embeddings** For 1D tokens (latent/state/env/lang), we use learned index embeddings $p_i \in \mathbb{R}^D$. For image tokens, we use 2D sinusoidal embeddings with coordinates normalized to $[0, 2\pi]$:

$$u_x(i) = 2\pi \frac{i}{W}, \quad u_y(j) = 2\pi \frac{j}{H}, \quad \omega_k = 10000^{\frac{2\lfloor k/2 \rfloor}{D/2}},$$

$$\text{PE}_x(2k) = \sin(u_x/\omega_k),\ \text{PE}_x(2k+1) = \cos(u_x/\omega_k),\ \text{PE}_y(2k) = \sin(u_y/\omega_k),\ \text{PE}_y(2k+1) = \cos(u_y/\omega_k),$$

and we concatenate $(\text{PE}_y, \text{PE}_x)$ channel-wise (as in code). Decoder slots use learned positional embeddings $P^{\text{dec}} \in \mathbb{R}^{S \times D}$.

**Transformer Encoder (Lightweight, Pre-Norm)**   We form
$$X = \text{concat}([x^{\text{lat}}, x^{\text{state}}, x^{\text{env}}, x^{\text{lang}}, X^{\text{img}}]) \in \mathbb{R}^{N \times D},$$
and corresponding $P^{\text{enc}} \in \mathbb{R}^{N \times D}$. Each encoder layer applies (pre-norm) multi-head self-attention (MHA) and a feed-forward network (FFN) with residuals:
$$\tilde{X} = X + \text{MHA}\big(\text{LN}(X)+P^{\text{enc}}, \text{LN}(X)+P^{\text{enc}}, \text{LN}(X)\big), \quad H = \tilde{X} + \text{FFN}\big(\text{LN}(\tilde{X})\big).$$
For a single head,
$$\text{Attn}(Q, K, V) = \text{softmax}\Big(\frac{QK^{\top}}{\sqrt{d_k}}\Big)V, \quad Q = (X+P^{\text{enc}})W_Q, \; K = (X+P^{\text{enc}})W_K, \; V = XW_V.$$
The encoder has $L_{\text{enc}}$ layers (4 in this paper) and ends with a LayerNorm.

**Transformer Decoder (Late Fusion via Cross-Attention)**   Decoder input is initialized as $Y^{(0)} = \mathbf{0} \in \mathbb{R}^{S \times D}$, and encoder output is denoted as $\mathcal{E}$. Each layer performs:

$$\textbf{(i) Self-attn:} \quad \hat{Y} = Y + \text{MHA}\big(\text{LN}(Y)+P^{\text{dec}}, \text{LN}(Y)+P^{\text{dec}}, \text{LN}(Y)\big),$$

$$\textbf{(ii) Cross-attn:} \quad \bar{Y} = \hat{Y} + \text{MHA}\big(\text{LN}(\hat{Y})+P^{\text{dec}}, \text{LN}(\mathcal{E})+P^{\text{enc}}, \text{LN}(\mathcal{E})\big),$$

$$\textbf{(iii) FFN:} \quad Y^{\text{out}} = \bar{Y} + \text{FFN}\big(\text{LN}(\bar{Y})\big).$$

Here cross-attention implements late fusion: queries come from decoder slots, while keys/values are the multi-modal encoder memory. We use dropout and either `relu`/`gelu` activations in the FFN, matching implementation.

**Action Head**   After $L_{\text{dec}}$ decoder layers and a final LayerNorm, each slot is mapped to action space with a linear head:
$$A = Y^{\text{final}}W_{\text{act}} + b_{\text{act}} \in \mathbb{R}^{H \times d_{\text{act}}}.$$
This corresponds to the per-timestep joint command (or other actuation) for a chunk of length $H$.

**Caching and Complexity**   Let $C_{\text{vis}}$ and $C_{\text{lang}}$ be the costs of frozen vision and language encoders, and $C_{\text{dec}}$ the decoder cost per step. Early fusion often recomputes both encoders each timestep: $T(C_{\text{vis}}+C_{\text{lang}}+C_{\text{dec}})$. Our decoupled late fusion caches language (and other static 1D tokens), yielding:
$$C_{\text{ours}} = T\,C_{\text{vis}} + C_{\text{lang}} + T\,C_{\text{dec}}.$$
Theoretical per-episode speedup is
$$\text{Speedup} = \frac{T(C_{\text{vis}}+C_{\text{lang}}+C_{\text{dec}})}{T\,C_{\text{vis}} + C_{\text{lang}} + T\,C_{\text{dec}}} > 1 \quad \text{for } T > 1.$$
Within attention, the dominant terms are $\mathcal{O}(N^2 D)$ in the encoder (small $L_{\text{enc}}$) and $\mathcal{O}(HND)$ in decoder cross-attention; both are lightweight due to small $SH$ and modest $N$ after flattening the final backbone stage.

**Variational Training Objective During Training**   Since our action chunking decoder predicts $H$ actions jointly. We followed traditional action chunking process Zhao et al. (2023) to add a low-dimensional latent $z$ per chunk turns the decoder into a conditional generative model $p_\theta(A_{1:H} \mid z, C)$ with context $C$ (vision, language, state), which produces $(\mu, \log \sigma^2)$ for $z$, using a CLS token head:
$$q_\phi(z \mid \text{state}, \text{actions}) = \mathcal{N}\big(z; \mu, \text{diag}(\sigma^2)\big), \quad \text{with } [\mu, \log \sigma^2] = W_{\text{post}}\, h_{\text{CLS}}.$$
The training loss combines action regression with a KL term:
$$\mathcal{L} = \underbrace{\frac{1}{S}\sum_{t=1}^{S} \|A_t - \hat{A}_t\|_2^2}_{\text{action loss}} + \beta\, D_{\text{KL}}(q_\phi(z) \,\|\, \mathcal{N}(0, I)).$$

This yields three benefits: (i) the KL term acts as an *information bottleneck* on $z$, encouraging a compact, chunk-level summary that regularizes the decoder; (ii) the stochastic $z$ enables *multi-modal* imitation (different valid action realizations) instead of regressing to means; and (iii) a single $z$ per chunk promotes *temporal coherence* across $t=1..H$.

At inference, we set $z=0$, so latency is identical to the non-variational path.

### A.3 Real Robot Tasks Setup for LeRobot

Here we provide the detailed task set up for LeRobot experiment, including language instruction and task goal. For Lerobot, we use the LeRobot SoArm-101 dual-arm system (one leader arm and one follower arm). We installed an Intel RealSense D435i RGB-D camera to provide a third-person view for the experiment. And in the experiment, we only use RGB images. We collect demonstration trajectories for 10 real-world tasks for training using LeRobot teleoperation. Each episode takes 9-18 seconds for the human operator to perform depending on the complexity of the task, which translates to 300-600 time steps given the control frequency of 30Hz. For each test trial, we randomize the initial location of the target object and repeat the experiment for 50 times to match the simulation set up. This experiment is especially challenging as the pre-training dataset (Bridge Data V2) did not include any rollouts from LeRobot, hence it will be harder for the model to predict the action for a new embodiment. We also attached the saved video roll outs for both real-robot experiment and simulation experiment in the supplementary experiment.

Tasks included in the finetuning dataset.

- **Pick-and-place Vitamin**: *Pick up the vitamin and place it on the tray.* The trial is counted as a success only when the robot correctly places the vitamin on the tray. This task tests the model's capability for handling slippy objects. The incorrect end-effector (gripper) placement could easily cause the vitamin slipped way and miss the target.

- **Pick-and-place Banana**: *Pick up the banana and place it on the tray.* The trial is counted as a success only when the robot correctly places the banana on the tray. Unlike previous papers using plastic toy vegetables or fruits, this task tests the model's capability for handling real deformable bananas. The incorrect gripper claw angle could squeeze the banana and cause the banana damaged.

- **Pick-and-place USB**: *Pick up the USB drive and place it on the tray.* The trial is counted as a success only when the robot correctly picks up the USB drive place it on the tray. This task tests the model's capability for picking up tiny objects.

- **Pick-and-place Spoon**: *Pick up the ice-cream spoon and place it on the tray.* The trial is counted as a success only when the robot correctly picks up the ice-cream spoon place it on the tray. This task tests the model's capability for picking up irregular shaped objects.

- **Pick-and-place Tissue**: *Pick up the tissue and place it on the tray.* The trial is counted as a success only when the robot correctly picks up the tissue place it on the tray. This task tests the model's capability for picking up tiny deformable objects.

- **Pick-and-place Juice Box**: *Pick up the juice box and place it on the tray.* The trial is counted as a success only when the robot correctly picks up the juice box and place it on the tray. This task tests the model's capability for picking up regular cube shaped objects.

- **Move towel**: *Move the towel from left to right hand side.* This experiment is counted as a success when the robot succeesfully grasp the edge of the cloth and move it to the right. This task tests the model's capability for handling thin slippy objects.

- **Open Lid**: *Open the lid and put on the tray.* This experiment is counted as a success when the robot succeesfully grasp the deformable lib handle and place it on the tray on the right hand side. This task tests the model's capability for precepting tiny handle and pick it up.

- **Close Lid**: *Pick up the lid from tray and close it.* This experiment is counted as a success when the robot successfully grasp the deformable lib handle from the tray and put it on the jar. This task is extremely challenge as it requires precise action generation to move the lid on top of the jar and make them fit seamlessly.

Unseen tasks for generalization on out-of-distribution objects and language instruction:

- **Move unseen towel**: *Move the towel from left to right hand side.* In this experiment, we use the same language instruction as the **Move Towel** task, but switch the test towel into a unseen one. This task is challenging as the new object is not seen in the finetuning dataset.

- **Pick-and-place hand cream**: *Pick up the hand cream and place it on the tray.* The trial is counted as a success only when the robot correctly picks up the hand cream and place it on the tray. This task tests the model's capability for understanding both unseen language instruction and objects.

## A.4 QUALITATIVE RESULTS ON SIMULATION ROLLOUTS

We present manipulation rollouts on four tasks from the LIBERO suite. Figures 8–11 qualitatively illustrate how NanoVLA recovers from common failure modes (e.g., mis-localization, slipped grasps, partial actuation) and ultimately completes long-horizon tasks. Across these examples, NanoVLA combines lightweight inference with robust error recovery.

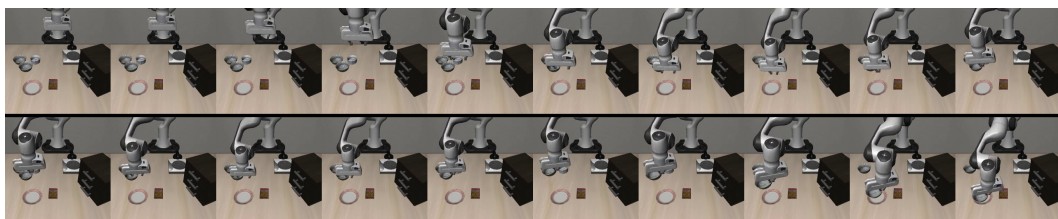

Figure 8: **LIBERO-spatial:** *Pick up the black bowl between the plate and the ramekin and place it on the plate.* Handling failed grasp to the bowl due to wrong object location estimation.

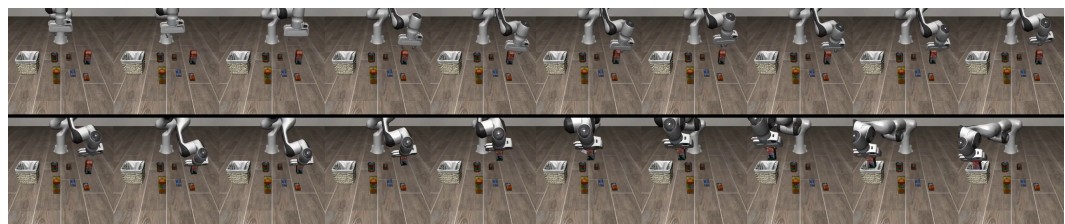

Figure 9: **LIBERO-object:** *Pick up the milk and place it in the basket.* Handling slipped grasp to the milk box due to wrong object location estimation.

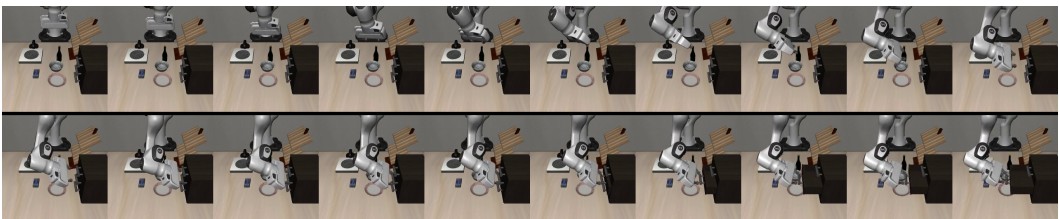

Figure 10: **LIBERO-goal:** *Open the middle drawer of the cabinet.* Handling stuck drawer.

## A.5 QUALITATIVE RESULTS ON REAL ROBOT ROLLOUTS

We further evaluate NanoVLA on real-robot manipulation. Consistent with our quantitative results, the model generalizes across diverse objects, environments, and language instructions, including previously unseen instances and prompts.

Figure 12 shows continuous pick-and-place under human intervention. The system monitors a region of interest and repeatedly executes the policy; after each successful transfer to the tray, an operator relocates the object to a random position. NanoVLA re-detects the new location and completes subsequent cycles reliably, despite perturbations.

In Figure 13 we compare rollouts on a challenging vitamin pick-up task. The container is round and slippery, making it prone to slipping or tipping under contact; precise action generation is required for a stable grasp. Thanks to its continuous-action decoder, NanoVLA produces more reliable grasps than SmolVLA in these real-world trials.

Figure 14 highlights out-of-distribution (OOD) generalization. The towel task demonstrates robustness to variations in appearance under a seen instruction. The hand-cream task demonstrates generalization to both an unseen object and an unseen instruction.

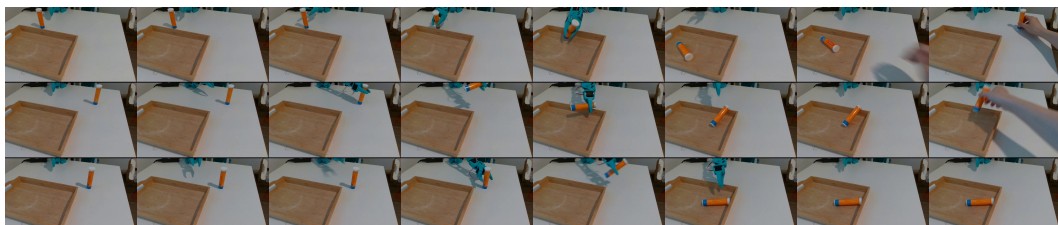

Figure 11: **LIBERO-long:** *Put both the alphabet soup and the tomato sauce in the basket.* Handling slipped grasp to the tomato sauce can due to wrong object location estimation.

Figure 12: **Pick-place multiple vitamin.** Continuous policy execution for random object locations using NanoVLA.

Finally, Figure 15 evaluates robustness across lighting conditions. Changing illumination alters shadows and color statistics, which can confound vision-language-action policies. Nevertheless, NanoVLA consistently localizes and manipulates the target across day and night settings.

## A.6 TRAINING GPU PERFORMANCE

We also evaluated the memory cost for NanoVLA and baseline, we launch the training job on 1 H20 GPU under varying batch sizes (1-32), and we measure the maximum GPU memory across training period (OpenVLA failed to run at batch 32). As shown in Figure 16a, results show that our proposed method consistently require less memory than SmolVLA due to lower trainable parameters.

Additionally, we also demonstrate the efficient training capability in Figure 16b on low computing resource device (e.g., entry level GPU RTX 3060, or edge computing device Jetson Orin Nano). Compares to existing VLAs at similar size (SmolVLA), NanoVLA achieves a significant reduction (~80% on RTX 3060, ~83% on Jetson) in training time, suggesting the effeiency of the proposed decoupling structure and caching mechanism. For reference, finetuning on LIBERO dataset usually takes 1M steps to converge, which only require 30 to 40 GPU hours on these edge devices, allowing a low-cost fine-tuning and potential on-device training and reinforcement learning.

## A.7 ABLATION STUDIES ON VISION BACKBONE

We provide additional ablation study on vision backbones to showcase the choice of ResNet-18 under constrained resources. Using ResNet-18 as our visual baseline (80.4% Avg.), scaling up the backbone yields only modest gains relative to the increase in parameters. Moving to ResNet-50 (~25M parameters vs. ~12M for ResNet-18) improves the average success rate to 81.7% (+1.3 points), while ResNet-101 (~45M parameters) further pushes the average to 83.1% (+2.7 points over ResNet-18), with most of the improvement concentrated on the more challenging Goal and Long tasks. However, these gains come with a roughly 2–4× increase in visual parameters and clear signs of diminishing returns (e.g., only +1.4 points from ResNet-50 to ResNet-101 for ~19M extra parameters). We therefore stop at ResNet-101 and do not adopt larger vision backbones such as ViT-B, which typically has on the order of 86M visual parameters: in our setting the overall model already contains about 0.5B parameters when combining the vision encoder, language backbone, and policy head, so adding a substantially heavier ViT-B backbone would significantly increase memory and latency with uncertain benefit on LIBERO-scale data.

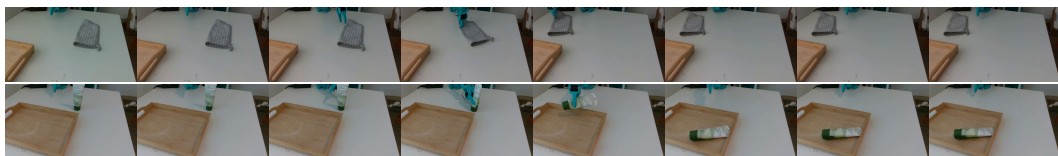

Figure 13: **Pick-place vitamin task.** (Above): SmolVLA rollout. (Below): NanoVLA rollout with visual trace.

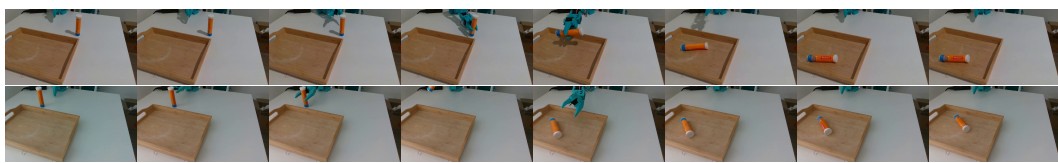

Figure 14: **OOD policy execution.** (Above): NanoVLA rollout for seen language instruction but unseen and object. (Below): NanoVLA rollout for unseen language instruction and object.

Figure 15: **Policy execution under different lighting condition.** (Above): NanoVLA rollout for policy execution during night. (Below): NanoVLA rollout for policy execution during day time.

### A.8 ABLATION STUDIES ON EFFICIENT VLAS

RT-1 is an extremely lightweight model at only 0.035B parameters, but it achieves just 0.16% success on LIBERO-90, indicating that in this setting its policy does not transfer meaningfully despite its efficiency. Our proposed NanoVLA-S, which has 0.161B parameters, outperform Rt-1 by 39% with additional 126M parameters, showing our proposed model's effective in balancing the trade off between model size the policy execution success rate. For larger models, such as SmolVLA (0.45B) and OpenVLA (7.5B), they achieve the success rate at 68.9% and 62.0%, respectively, showing that larger VLA models can leverage their capacity for substantially better multi-task performance. In contrast, our NanoVLA variants occupy a more favorable part of the accuracy–efficiency trade-off: NanoVLA-S (0.161B) is smaller than SmolVLA yet remains competitive at 55.1%, while NanoVLA-L (0.52B) and NanoVLA-R (0.296B) achieve 83.3% and 81.6%, respectively. NanoVLA-L is only slightly larger than SmolVLA (0.52B vs. 0.45B) but improves performance by +14.4 points, and it even outperforms the 7.5B-parameter OpenVLA by +21.3 points, despite being over an order of magnitude smaller. These results show that, while ultra-compact policies like RT-1 are attractive from a parameter standpoint, they are insufficient for LIBERO-90, and our NanoVLA design attains substantially better performance than both similarly sized and much larger VLAs, offering a strong balance between efficiency and capability.

### A.9 IMPLEMENTATION DETAILS

We provide the detailed implementation for NanoVLA in Table 6 and the full code base can be found on `https://anonymous.4open.science/r/nanovla-38EC`.

Additionally, for Jetson Nano deployment, we can follow the following steps. We faced some kernel issues that prevented us from installing the CUDA-supported PyTorch package on the device directly. Instead, we leverage the pre-built LeRobot Jetson container with CUDA support as a hack to install NanoVLA's dependencies.

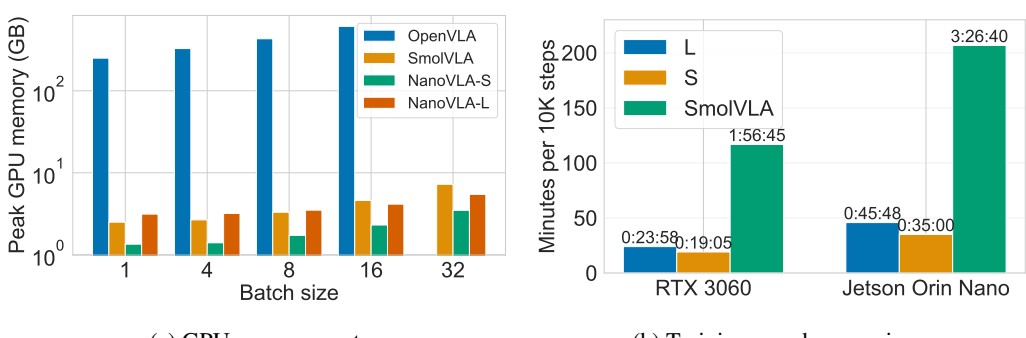

(a) GPU memory cost.          (b) Training speed comparison.

Figure 16: Computational efficiency analysis on Jetson Orin Nano.

| Visual | Spatial | Object | Goal | Long | Avg. |
|---|---|---|---|---|---|
| ResNet-18 | 87.2 | 89.8 | 90.0 | 55.2 | 80.4 |
| ResNet-50 | 87.4 | 89.2 | 92.8 | 57.4 | 81.7 |
| ResNet-101 | 87.4 | 92.0 | 94.4 | 58.6 | 83.1 |

Table 4: Performance (%) of different visual backbones on Spatial, Object, Goal, and Long tasks.

## JETSON-CONTAINERS SETUP

```
1  git clone https://github.com/dusty-nv/jetson-containers
2  bash jetson-containers/install.sh
3  cd jetson-containers
4  ./packages/robots/lerobot/clone_lerobot_dir_under_data.sh
5  cp -r <NanoVLA repo location>
   ↪   /home/peter/jetson-containers/data/lerobot
6  ./packages/robots/lerobot/copy_overlay_files_in_data_lerobot.sh
7  ./run.sh -v ${PWD}/data/lerobot/:/opt/lerobot/ $(./autotag lerobot)
```

## JETSON EVALUATION

```
1  # NanoVLA installation
2  pip install --isolated -e .
3  pip install --isolated -r requirements.txt
4  # LIBERO Evaluation
5  cd LIBERO/
6  pip install -e .
7  cd ..
8  pip install -r src/lerobot/libero/libero_requirements.txt
9
10 TOKENIZERS_PARALLELISM=false python -m
   ↪   lerobot.libero.lerobot_inference \
11   --policy_path=<trained policy> \
12   --task_suite_name=libero_spatial \
13   --act_len=10
```

| Model | RT-1 | SmolVLA | OpenVLA | NanoVLA-S | NanoVLA-L | NanoVLA-R |
|---|---|---|---|---|---|---|
| #param. | 0.035B | 0.45B | 7.5B | 0.161B | 0.52B | 0.296B |
| SR | 16.1 | 68.9 | 62.0 | 55.1 | **83.3** | 81.6 |

Table 5: Efficient VLAs' performance on LIBERO-90 benchmark (%).

| Component | Hyperparameter | Value |
|---|---|---|
| Training & Optimization | | |
| | Global steps | 1,000,000 |
| | Batch size | 8 |
| | Seed | 1000 |
| | Optimizer | AdamW |
| | Learning rate | $1 \times 10^{-5}$ |
| | Weight decay | $1 \times 10^{-4}$ |
| | Grad clip norm | 10.0 |
| | Scheduler | None |
| | Mixed precision | No |
| | Num workers | 4 |
| | Checkpoint every (steps) | 100,000 |
| Dataset & Preprocessing | | |
| | Use ImageNet stats | Yes |
| | Image transforms | Disabled (max_num_transforms=3, random_order=No) |
| Policy / Architecture | | |
| | Type | `nanovla` |
| | Use language | [Yes (`qwen`) │ No (`bert`)], frozen LM = Yes |
| | Vision backbone | ResNet-18 (pretrained = `IMAGENET1K_V1`) |
| | dim_model | 512 |
| | n_heads | 8 |
| | n_encoder_layers | 4 |
| | n_decoder_layers | 1 |
| | Activation | ReLU |
| | Dropout | 0.1 |
| | Chunk size | 100 |
| | n_obs_steps | 1 |
| | n_action_steps | 100 |
| | Normalization | ACTION = mean/std; STATE = mean/std; VISUAL = identity |
| | Policy optimizer | lr = $1 \times 10^{-5}$; backbone lr = $1 \times 10^{-5}$; wd = $1 \times 10^{-4}$ |
| Evaluation | | |
| | Eval episodes | 50 |

Table 6: Hyperparameters used for the `NanoVLA` policy.

