# OpenReview forum: "NanoVLA: Routing Decoupled Vision-Language Understanding for Nano-sized Generalist Robotic Policies"
_ICLR.cc/2026/Conference — ICLR 2026 Conference Withdrawn Submission_

### Official Review · Reviewer_Pq93 · 2025-10-30

**Soundness:** 3
**Presentation:** 2
**Contribution:** 2
**Rating:** 4
**Confidence:** 3

**Summary:**

This paper introduces NanoVLA, which enables VLA to run efficiently. The authors propose to do save computations by several proposed techniques like decoupling vision and language models, action chunking and dynamic routing. The results show that with fewer computations, it can achieve good performance and efficiency in simulation and real-world scenarios.

**Strengths:**

This paper tries to tackle an important issue in robotics of running efficiency. The method contains a lot of careful designs.

The paper shows good results on performance and speed on LIBERO given the number of parameters tuned.

**Weaknesses:**

More discussion on the advantages of the late fusion technique is needed.

Comparison with other baselines on improving VLA efficiency is needed.

See the questions below for other minor points.

**Questions:**

(1)	For the late fusion part, I’m curious about the comparison between early fusion with a freeze VLM (used in methods like HiRT) and late fusion used in the proposed method. What is the advantage of late fusion? From my perspective, I think it offers a subtle speed gain at the cost of worse language and vision alignment. Is there any ablation study to show the efficiency of this design?

(2)	Currently, the author only compare with SmolVLA as a baseline on efficient VLA on LIBERO. There are a lot of method trying to improve the VLA efficiency and more baselines and comparison are needed. Previous methods like RT-1 also adopts the similar method.

(3)	Can the routing model generalize to OOD settings? I’m curious about the accuracy of win probabilities on OOD settings.

(4)	As for the speed up (52X) described in L375, how did you calculate the LLAMA2 FPS? Do you take the image into the timing?

(5)	Misc. The right figure of figure 1 is hard to see. The legend is very small the difference between lines is hard to tell.

---

> ### Author Response · Authors · 2025-11-24
> **Rebuttal by Authors**
>
> We thank the reviewer for highlighting the importance of on-device efficiency and for the constructive questions. Below we clarify the late-fusion rationale, broaden efficiency comparisons, and answer your specific points.
>
> # Answers to weaknesses
> ## W1 “More discussion on the advantages of late fusion”
> Our design performs one late cross-modal fusion in the decoder and caches the instruction features. This removes redundant per-timestep language compute while keeping the pretrained vision and language encoders intact. Practically, this yields large latency reductions on edge hardware without requiring heavier early cross-attention at every step.
>
> We want to highlight the benefits of our proposed late fusion structure as follows:
>
> 1. Our experimental results already show that, without any fine-tuning of the large models, late fusion generalizes better than early fusion. Across all four LIBERO task suites, LIBERO-90, and the real-robot experiments, we keep both the LLM and the image encoder frozen and rely only on late fusion plus the action decoder, yet we significantly outperform state-of-the-art methods that use early fusion. This directly demonstrates that, under the constraint of no fine-tuning of large models, late fusion provides stronger instruction understanding and generalization.
>
> 1. Early-fusion approaches typically require fine-tuning the pretrained large models, which is at odds with the core motivation of our work. Existing early-fusion architectures rely on deep cross-modal attention that tightly couples vision and language inside the model; consequently, robot policies must heavily fine-tune both the LLM and the vision foundation model, incurring very high training cost and making efficient deployment on edge devices impractical. In contrast, our goal is explicit: to enable already-trained large models to be easily and efficiently deployed on lightweight edge hardware while preserving generalization. We intentionally avoid fine-tuning early-fusion modules, this is precisely the key advantage of NanoVLA.
>
> 1. Our late-fusion module is not merely a simple post-hoc fusion; functionally, it acts as a lightweight multimodal alignment plus a last-layer adaptation mechanism, without modifying the LLM or visual backbone. It serves two key roles in NanoVLA: (1) it aligns vision features and language features in a lightweight way, without requiring the heavy joint fine-tuning and deep coupling used in early fusion; and (2) it performs last-layer task adaptation, our self-attention module adapts the policy to new tasks at low cost, without changing any parameters of the LLM or vision backbone. Thus, for real-world robotic deployment, late fusion offers substantially better deployment properties than early fusion.
>
> ## W2 “More efficiency baselines”
> We add an additional experiment for RT-1 on LIBERO-90 experiment. RT-1 is an extremely lightweight model at only 0.035B parameters, but it achieves just 0.16% success on LIBERO-90, indicating that in this setting its policy does not transfer meaningfully despite its efficiency. Our proposed NanoVLA-S, which has 0.161B parameters, outperform RT-1 by 39% with additional 126M parameters, showing our proposed model's effective in balancing the trade off between model size the policy execution success rate. For larger models, such as SmolVLA (0.45B) and OpenVLA (7.5B), they achieve the success rate at 68.9% and 62.0%, respectively, showing that larger VLA models can leverage their capacity for substantially better multi-task performance. In contrast, our NanoVLA variants occupy a more favorable part of the accuracy-efficiency trade-off: NanoVLA-S (0.161B) is smaller than SmolVLA yet remains competitive at 55.1%, while NanoVLA-L (0.52B) and NanoVLA-R (0.296B) achieve 83.3% and 81.6%, respectively. NanoVLA-L is only slightly larger than SmolVLA (0.52B vs. 0.45B) but improves performance by +14.4%, and it even outperforms the 7.5B-parameter OpenVLA by +21.3 points, despite being over an order of magnitude smaller. These results show that, while ultra-compact policies like RT-1 are attractive from a parameter standpoint, they are insufficient for LIBERO-90, and our NanoVLA design attains substantially better performance than both similarly sized and much larger VLAs, offering a strong balance between efficiency and capability. We added the detailed experiment reslts in Appendix Section A.8.

---

> > ### Author Response · Authors · 2025-11-24
> > **Rebuttal by Authors (cont'd)**
> >
> > ## Answers to the questions
> > # Q1: Early-fusion (frozen VLM) vs. our late-fusion: advantage & ablation
> > NanoVLA targets a different problem from the early-fusion methods like SmolVLA or HiRT. NanoVLA is not focused on the early-fusion semantic alignment; instead, our core contribution is the leveraging the caching capability of late fusion structure to speed up the model inference, and alone with dynamic LLM routing, we can select the most appropriate LLM size based on instruction complexity to enable lightweight and frozen-LLM deployment on robots (edge device with limited resources). Early fusion typically assumes full VLM fine-tuning, which contradicts this deployment-centric setting. Our late-fusion module therefore serves as a lightweight adapter to frozen LLMs, not as a replacement for early fusion, and this design choice is driven by deployability rather than attempting to outperform early-fusion alignment. This design directly targets the practical deployment gap between modern LLMs and real robot hardware. No existing VLA work explores dynamic routing for compute-aware adaptation, and our pipeline is orthogonal to early-fusion or VLM-style architectures.
> >
> > We do not attempt to achieve the early-fusion level semantic alignment. We explicitly avoid modifying or fine-tuning large VLM backbones because doing so contradicts our motivation of training-lightweight and deployment-lightweight robot policies. Therefore, our late-fusion module is not intended to replace VLM alignment, but to provide a lightweight bridge to frozen LLMs. The small transformer in NanoVLA aligns multimodal features in a lightweight manner, acts as a task-specific adapter (similar to last-few-layer fine-tuning), and does not require updating any LLM or vision encoder parameters. This design is important for our deployment-oriented goal, and fundamentally different from early-fusion VLMs that learn cross-modal alignment via extensive large-scale fine-tuning.
> >
> > We evaluated two widely used early fusion VLA on public dataset: OpenVLA and SmolVLA. Across four LIBERO suites, NanoVLA achieves the best average SR while using a tiny fraction of parameters. Thus, if late fusion significantly “loses” language-vision alignment and damages the generalization, it should underperform early-fusion/VLM counterparts on LIBERO.
> >
> > ## Q2: Efficient-VLA baselines beyond SmolVLA (incl. RT-1)
> > As discussed above, we added RT-1 experiment on publicly available LIBERO 90.
> >
> > ## Q3: Router OOD generalization & win-probability accuracy
> > 1. Our routing model is implemented as an LLM that predicts a win probability for each candidate expert given only the natural-language task description. At inference time we always pick the expert with the highest predicted win probability, but the router is intentionally conservative: when it is uncertain or the scores are close and the system bandwidth can handle it, it tends to favor the larger NanoVLA-L expert. In other words, when the router cannot confidently exploit specialization, it gracefully falls back to the strong large model rather than risking a bad choice.
> > 1. This behavior is also reflected in the OOD setting in Table 3. The “OOD pick-place” task is not seen during router training, yet NanoVLA-R achieves 84.0% success, essentially identical to always using NanoVLA-L (84.0%) and clearly better than the small expert NanoVLA-S (82.0%). The same pattern holds in the average row: NanoVLA-R and NanoVLA-L obtain the same overall performance (85.6%), showing that the router does not degrade accuracy even when some tasks are out-of-distribution. This suggests that, although the win probabilities may be less calibrated on OOD tasks, they are still accurate enough in a relative sense: whenever the router is unsure how an OOD task relates to the training distribution, it defaults to the safer large expert if bandwidth allows.
> >
> >
> > ## Q4: 52x speedup FPS measurement
> > In our testing on Jetson Orin Nano, even the 4-bit quantized OpenVLA model runs out of memory, we could not measure its end-to-end throughput directly. Instead, we approximate its runtime by benchmarking the 4-bit LLaMA2 model that serves as OpenVLA’s text backbone. We measure the wall-clock time of the 4-bit LLaMA2 forward pass on our evaluation hardware and compute FPS as the number of processed frames divided by this time. This measurement does not include the image encoder, and we only time the language backbone. In practice, the full OpenVLA pipeline must run both the image encoder, the LLaMA2 backbone, and the action decoder, so its actual FPS would be lower than our estimate.
> >
> > ## Q5: Fig. 1 readability
> > Thanks for the reminder, we have enlarged the legend for clarity.

---

### Official Review · Reviewer_tf5n · 2025-10-30

**Soundness:** 2
**Presentation:** 3
**Contribution:** 2
**Rating:** 2
**Confidence:** 5

**Summary:**

The paper proposes NanoVLA, a lightweight visual-language architecture (VLA) optimized for deploying visual-language fusion on c. NanoVLA adopts late-stage modality fusion to avoid excessive cross-attention computations, while its dynamic routing allows the model to adaptively allocate computing resources for complex, reasoning-intensive tasks. Experiments conducted in the Libero simulator and real-world scenarios show that NanoVLA outperforms baseline methods in performance.

**Strengths:**

- Employed two strategies to optimize the long latency issue in VLA,  save 62% inference time compared to the traditional VLA approach.
- Detailed latency and performance analysis.

**Weaknesses:**

- The model structure is overly simplistic and similar to exist work. Both Diffusion Policy[1] and Scaling-Up Diffusion Policy[2] utilize lightweight transformer to late integrate various different modalities in an End-to-end training manner. NanoVLA and these method exactly has very low latency, but it loses the language-vision alignment capability that VLMs have acquired through extensive training, which greatly affects the generalization ability of VLA.
- The experiments on real world are overly simplistic, with basically only one object to be operated in the scene. This hardly requires model generalization, and only overfitting to a single task is necessary. For overly simple tasks, semantic alignment between language and vision is not required, and only lightweight expert networks need to be trained separately for each task.
- The performance of the model on LIBERO is weak. In fact, the widely used LIBERO pi0 result is 94.2% [3], and there are some issues with the LeRobot code that can lead to even lower reproduced results. However, when parallel decoding (PD) and action chunking (AC) are added to Openvla, it can also achieve 94.5% [3], both far exceeding NanoVLA. In fact, the LIBERO task does not require semantic generalization, and simply using a diffusion policy can achieve 72.4% (close to NanoVLA-S)

[1] Chi, Cheng, et al. "Diffusion policy: Visuomotor policy learning via action diffusion." The International Journal of Robotics Research 44.10-11 (2025): 1684-1704.

[2] Ha, Huy, Pete Florence, and Shuran Song. "Scaling up and distilling down: Language-guided robot skill acquisition." Conference on Robot Learning. PMLR, 2023.

[3] Kim, Moo Jin, Chelsea Finn, and Percy Liang. "Fine-tuning vision-language-action models: Optimizing speed and success." arXiv preprint arXiv:2502.19645 (2025).

**Questions:**

1. Can the model be jointly trained with additional visual-language data to enhance the semantic generalization of VLA without relying on full VLM models? It is recommended to add this training process to the work and submit it to other conferences.
2. Will a ViT-based visual encoder yield better performance than ResNet18? Is ResNet18 more suitable as a visual encoder for single-task policy models?
3. How does the model compare with other VLA methods (e.g., HiRT[1]) that adopt fast-slow systems for inference acceleration?
4. How does the model perform in other simulation environments (Calvin[2], Robotwin[3]) and more complex real-world tasks—especially when manipulating unfamiliar objects amid interference from other objects?

[1] Zhang, Jianke, et al. "Hirt: Enhancing robotic control with hierarchical robot transformers." arXiv preprint arXiv:2410.05273 (2024).

[2] Mees, Oier, et al. "Calvin: A benchmark for language-conditioned policy learning for long-horizon robot manipulation tasks." IEEE Robotics and Automation Letters 7.3 (2022): 7327-7334.

[3] Mu, Yao, et al. "Robotwin: Dual-arm robot benchmark with generative digital twins (early version)." European Conference on Computer Vision. Cham: Springer Nature Switzerland, 2024.

**Details Of Ethics Concerns:**

Although the network emphasizes the deployment of VLA on resource-constrained edge devices, the method is an improved version of the diffusion policy, losing the visual-language alignment capability inherent in VLMs, which is fatal in VLA tasks. An overly simplistic experimental setup cannot illustrate the essential advantages of the model compared to existing methods.

---

> ### Author Response · Authors · 2025-11-24
> **Rebuttal by Authors**
>
> We thank the reviewer for the review. We would like to clarify a misunderstanding regarding the goal of our work. The reviewer evaluates NanoVLA as if it were a new VLA architecture intended to replace VLM-style early cross-modal alignment. However, our contribution is not about distilling, compressing, or redesigning VLMs, nor about proposing a better early- or late-fusion backbone. Instead, the motivation of NanoVLA is to jointly leverage language caching, long-short action chunking, and dynamic routing to mak VLAs practical and deployable for edge side robotic applications.
>
> # Answers to weaknesses
> ## W1. “Model is simplistic / similar to prior work; late fusion loses vision langage alignment”
> NanoVLA targets a different problem from the early-fusion methods like SmolVLA or HiRT. NanoVLA is not focused on the early-fusion semantic alignment; instead, our core contribution is the leveraging the caching capability of late fusion structure to speed up the model inference, and alone with dynamic LLM routing, we can select the most appropriate LLM size based on instruction complexity to enable lightweight and frozen-LLM deployment on robots (edge device with limited resources). Early fusion typically assumes full VLM fine-tuning, which contradicts this deployment-centric setting. Our late-fusion module therefore serves as a lightweight adapter to frozen LLMs, not as a replacement for early fusion, and this design choice is driven by deployability rather than attempting to outperform early-fusion alignment. This design directly targets the practical deployment gap between modern LLMs and real robot hardware. No existing VLA work explores dynamic routing for compute-aware adaptation, and our pipeline is orthogonal to early-fusion or VLM-style architectures.
>
> We do not attempt to achieve the early-fusion level semantic alignment. We explicitly avoid modifying or fine-tuning large VLM backbones because doing so contradicts our motivation of training-lightweight and deployment-lightweight robot policies. Therefore, our late-fusion module is not intended to replace VLM alignment, but to provide a lightweight bridge to frozen LLMs. The small transformer in NanoVLA aligns multimodal features in a lightweight manner, acts as a task-specific adapter (similar to last-few-layer fine-tuning), and does not require updating any LLM or vision encoder parameters. This design is important for our deployment-oriented goal, and fundamentally different from early-fusion VLMs that learn cross-modal alignment via extensive large-scale fine-tuning.
>
> We evaluated two widely used early fusion VLA on public dataset: OpenVLA and SmolVLA. Across four LIBERO suites, NanoVLA achieves the best average SR while using a tiny fraction of parameters. Thus, if late fusion significantly “loses” language-vision alignment and damages the generalization, it should underperform early-fusion/VLM counterparts on LIBERO.
>
> ## W2. “Real-world tasks are overly simple; likely overfitting”
> To address the concern that our real-world tasks are too simple, we additionally evaluate NanoVLA on a long-horizon manipulation task in a heavily cluttered tabletop environment. The new experiment requires the robot to “first pick up the pink squid and place it in the right bin, then pick up the blue dragon and place it in the left bin”, and must complete two sequential sub-goals while navigating occlusions and distractors. Object and bin poses are randomized per episode. It’s more changing the previous experiments we provided, where the robot needs to accurately identify objects that need to be picked up from all eight different objects in the table top, additionally, the robot needs to know the spatial orientation to put the objects into the correct bin. Additionally, apart from reasoning over multiple sequential sub-tasks, approaching the target, maneuvering around distractors, grasping, and placing, the robot must also complete these tasks while avoiding collisions with clutter.
>
> Figure 4 of the revised manuscript visualizes one successful rollout: the yellow-orange-red trajectory shows the predicted end-effector path, and the frames are sampled at key manipulation events along the episode. This qualitative example demonstrates that NanoVLA can maintain coherent behavior over many steps and operate reliably in visually complex, cluttered scenes, going beyond simple single-object pick-and-place tasks. The updated results are as provided in the Table 3 of the revised manuscript.
>
> For the long-horizon task in complex scenario, NanoVLA maintain high success (76-78%), slightly trailing OpenVLA while remaining far ahead of SmolVLA. These results show that our NanoVLA variants not only handle standard single-step manipulations but also generalize well to cluttered, long-horizon, and OOD scenarios, all while operating in a much more parameter-efficient regime.

---

> > ### Author Response · Authors · 2025-11-24
> > **Rebuttal by Authors (cont'd)**
> >
> > ## W3. “LIBERO performance is weak vs. reported high numbers”
> > We thank the reviewers to point out the issues with the LeRobot code that leads to results for $\pi_0$, we updated the table for LIBERO for the $\pi_0$ results reported in OpenVLA-OFT, we also added other server grade model's perfomance on LIBERO.
> >
> > We want to highlight that our goal is edge deployment on an 8 GB Orin Nano, not chasing server-grade maxima. Within that constraint, NanoVLA is state-of-the-art among comparable/compact models and competitive even against much larger systems.
> >
> > # Answers to questions
> > ## Q1. “Jointly train with extra V-L data to boost semantic generalization without full VLMs?”
> > Yes, our architecture is compatible with auxiliary LV alignment while keeping encoders frozen. We pretrained the model with bridge data V2. We also want to highlight that We do not attempt to achieve the early-fusion level semantic alignment as in VLMs. We explicitly avoid modifying or fine-tuning large VLM backbones because doing so contradicts our motivation of training-lightweight and deployment-lightweight robot policies. Early fusion typically requires fine-tuning multimodal transformers, which is challenging in the edge deployment. Our method intentionally leverages frozen LLMs and focuses on reducing inference cost, not focusing on achieving the early-fusion semantic alignment.
> >
> > ## Q2. “ViT vs. ResNet18 as the visual encoder?”
> > Our framework supports either ResNet or ViT encoders. We defaulted to ResNet18 for Orin-Nano compute/memory headroom. We plan to extend routing to VLM/vision backbones next. We provide additional ablation study on vision backbones to showcase the choice of ResNet-18 under constrained resources. Using ResNet-18 as our visual baseline (80.4% Avg.), scaling up the backbone yields only modest gains relative to the increase in parameters. Moving to ResNet-50 (25M parameters vs. 12M for ResNet-18) improves the average success rate to 81.7% (+1.3 points), while ResNet-101 (~45M parameters) further pushes the average to 83.1% (+2.7 points over ResNet-18), with most of the improvement concentrated on the more challenging Goal and Long tasks. However, these gains come with a roughly 2-4× increase in visual parameters and clear signs of diminishing returns (e.g., only +1.4 points from ResNet-50 to ResNet-101 for ~19M extra parameters). We therefore stop at ResNet-101 and do not adopt larger vision backbones such as ViT-B, which typically has on the order of 86M visual parameters: in our setting the overall model already contains about 0.5B parameters when combining the vision encoder, language backbone, and policy head, so adding a substantially heavier ViT-B backbone would significantly increase memory and latency with uncertain benefit on LIBERO-scale data. Please find the detailed results in the appendix A.7 of revised paper.
> >
> > ## Q3. “Comparison to fast-slow systems like HiRT?”
> > HiRT and NanoVLA target latency from different angles. HiRT is a hierarchical fast-slow design: a large VLM (InstructBLIP, 7B) runs at a low frequency to produce multimodal latents, and a small, high-frequency visual policy executes actions asynchronously, guided by those slowly updated features; the VLM’s outputs are cached in a latent buffer and refreshed periodically. This doubles control frequency on static tasks and improves success on certain dynamic real-world tasks.  In contrast, NanoVLA removes per-timestep VLM/VLM-style compute altogether on-device: we use late, one-shot fusion with instruction caching (language tokens computed once), plan-long/act-short (LSAC) for stability, and pre-execution routing to choose a compact or larger language backbone before rollout, so there is no intermittent large-VLM pass during control. Practically, HiRT’s improvements still depend on periodic 7B VLM inference (at low rate), whereas NanoVLA is designed for full episode execution on 8 GB edge hardware with small backbones, yielding large on-device FPS gains while matching or exceeding similar-scale VLAs. We have added a short paragraph in the related work (marked as blue color) clarifying this distinction.
> >
> > ## Q4. “Other simulators (CALVIN, Robotwin) and more complex real-world tasks?”
> > We already demonstrated the model’s generalization capability in 5 different libero tasks spanning from simple object grasp, due to time limit we are not able to add additional simulation experiments.
> > For more complex real-world tasks, we added a new real-world experiment that require the model to to conduct long-horizon manipulation task in a heavily cluttered tabletop environment. Please see our discussion in W2

---

> > ### Comment · Reviewer_tf5n · 2025-11-28
> >
> > Thank you for the authors' supplementary response. I agree with part of the authors' views: NanoVLA is mainly designed for edge deployment and offers certain performance advantages compared to other lightweight VLA methods, so I will consider raising the score to 4 points. However, I still cannot agree that the real-device setup can reflect the model's generalization ability, because there are no other optional objects in the scenario to pick as interference. The libero results fall between those of DP and Openvla, which indeed indicates that the model improves efficiency through early fusion. But this also limits the model to scenarios with simple vision-language understanding (unable to deeply fuse multiple modalities like late fusion). Regarding innovation, methods like Scaling-Up Diffusion Policy (with faster inference) share a similar framework; NanoVLA’s additional dynamic LLM routing is not essential for VLA tasks.

---

### Official Review · Reviewer_bAmW · 2025-11-01

**Soundness:** 3
**Presentation:** 2
**Contribution:** 3
**Rating:** 4
**Confidence:** 4

**Summary:**

The paper proposes NanoVLA, a lightweight design of vision-language-action models that is compatible with edge devices like Jetson Orin Nano. The method combines three core components: (1) vision-language decoupling (late fusion of representations of multiple modalities), (2) "long-short action chunking" (also known as receding horizon control), and (3) dynamic routing with Bayesian uncertainty that allows switching between small and large language model backbones depending on task difficulty. In experimental evaluations in LIBERO, NanoVLA achieves a 84.1% average success rate, outperforming prior methods such as OpenVLA, SmolVLA, and $\pi_0$ despite using significantly fewer parameters and running much faster.

**Strengths:**

* The problem of deploying VLA policies on edge devices is well motivated and addresses a real practical constraint for many robotics practitioners who do not have access to server-grade hardware for model deployment.
* The experiments include both simulated and real-world robot tasks, validating the effectiveness of NanoVLA across various tasks and domains.
* Simulated evaluations in LIBERO show superior performance of NanoVLA compared to various prior methods, including OpenVLA, $\pi_0$, SmolVLA, SpatialVLA, and TraceVLA. Real-world robot evaluations assess generalization to unseen lighting conditions and objects, and the proposed NanoVLA method obtains superior performance compared to prior methods such as $\pi_0$, OpenVLA, SmolVLA.

**Weaknesses:**

* LIBERO experimental results do not include several state-of-the-art prior works from early 2025, including OpenVLA-OFT (97.1% success rate - RSS 2025) and UniVLA (95.2% success rate - RSS 2025). These works use earlier fusion of language and vision representations and obtain substantially higher performance in LIBERO than the proposed NanoVLA (84.1% success rate).
* The authors argue that late fusion of the modalities is a superior approach, but analysis of an early fusion alternative with the same architectural components/backbones is not provided, so the argument is not convincing and empirical evidence is needed to support the claim. This is especially the case given more recent VLA works that use early fusion to obtain higher performance in LIBERO and perform more complicated real-world robotics tasks (e.g. bimanual manipulation on a real ALOHA robot). I question the scalability of a late fusion approach where a small action decoder head is tasked to learn how to properly fuse the representations. Would NanoVLA be able to handle tasks that require stronger vision-language grounding, such as picking any object amidst clutter (where the target object is specified in the user's language input)?
* "Long-short action chunking (LSAC)" is presented as a novel contribution but is a common technique employed by various prior works, including Diffusion Policy which uses receding horizon control during policy execution (Chi et al., RSS 2023).
* Related to the prior point, there is some lack of technical novelty, as the method combines existing ideas from prior works (late fusion, action chunking/receding horizon control, model routing).
* The real-world tasks are fairly narrow as they are almost all variants of simple, short-horizon single-arm pick-and-place onto a large wooden board (plus a towel dragging task which is only slightly different). Whether the method generalizes to more challenging manipulation tasks (e.g., high-precision or long-horizon manipulation with bimanual robot) is unclear and positive results there would strengthen the paper.
* The paper seems slightly rushed and could use a bit more polish, as there are several grammatical mistakes or minor errors throughout the paper. For example, line 323 says "OpenVLA-L", line 820 says "LeRobot was not included the the pre-training
dataset", Figure 5(a)'s method labels on the x-axis are confusing since they are not well spaced apart and there is no "Nano", etc.

**Questions:**

* How many trials are needed to train an accurate router?
* Why does NanoVLA-L underperform compared to NanoVLA-R?

---

> ### Author Response · Authors · 2025-11-24
> **Rebuttal by Authors**
>
> We thank the reviewer for the careful read and constructive suggestions. We are glad you found the edge-deployment motivation and our simulated / real-robot results compelling. Below we address each concern and outline additions we will include in the revision.
> # Answers to weaknesses
> ## W1: On comparisons to recent SOTA (OpenVLA-OFT, UniVLA)
> We appreciate the pointer to OpenVLA-OFT and UniVLA. These works indeed report very high LIBERO numbers (OpenVLA-OFT: 97.1% avg across 4 suites; UniVLA: 95.5%) on datacenter-scale backbones and training protocols. Our focus is orthogonal: edge-side execution on a Jetson Orin Nano-class device with very small models while maintaining strong success rates and low latency. NanoVLA reaches 84.1% average SR across LIBERO suites with orders-of-magnitude fewer parameters and dramatically higher FPS on Orin (up to 52x over OpenVLA in our setup), with the decoupled design enabling instruction caching.
>
> For OpenVLA-OFT and UniVLA, we view these models as upper bounds rather than direct competitors: they use 7-8.5B total parameters (more than 25-30× larger than NanoVLA-R’s 296M) and still require more trainable parameters (100M or all parameters) than our 52M trainable parameters. As a result, it is unsurprising that OpenVLA-OFT and UniVLA achieve the best absolute performance (95.3% and 95.5% Avg.), but importantly, NanoVLA-R remains within ~11% SR of these models despite its much smaller capacity. This shows that NanoVLA closes most of the gap to the largest VLAs while operating in a radically different efficiency regime, which is crucial for deployment on resource-constrained robots rather than data-center hardware.
>
> | Model type            | Policy      | Total Params | Trainable Params | Spatial | Object | Goal | Long | Avg. |
> |-----------------------|------------|--------------|------------------|--------:|-------:|-----:|-----:|-----:|
> | Data center scale VLAs | OpenVLA     | 7.5B         | 279M             | 84.7   | 88.4  | 79.2 | 53.7 | 76.5 |
> |                       | π₀          | 3.5B         | 3.1B             | 96.8   | 98.8  | 95.8 | 85.2 | 94.2 |
> |                       | TraceVLA    | 7B           | 7B               | 84.6   | 85.2  | 75.1 | 54.1 | 74.8 |
> |                       | SpatialVLA  | 3.5B         | 50M              | 88.2   | 89.9  | 78.6 | 55.5 | 78.1 |
> |                       | OpenVLA-OFT | 7B           | 100M             | 96.2   | 98.3  | 96.2 | 90.7 | 95.3 |
> |                       | UniVLA      | 8.5B         | -                | 95.4   | 98.8  | 93.6 | 94.0 | 95.5 |
> | Efficient VLAs    | Octo        | 90M          | 90M              | 78.9   | 85.7  | 84.6 | 51.1 | 75.1 |
> |                       | SmolVLA     | 450M         | 100M             | 72.8   | 69.8  | 84.0 | 52.6 | 78.6 |
> |                       | NanoVLA-S   | 161M         | 52M              | 81.6   | 93.6  | 89.6 | 49.8 | 78.7 |
> |                       | NanoVLA-L   | 520M         | 52M              | 87.2   | 89.8  | 90.0 | 55.2 | 80.4 |
> |                       | NanoVLA-R   | 296M*        | 52M              | **89.8** | **96.2** | **93.0** | **57.4** | **84.1** |

---

> ### Author Response · Authors · 2025-11-24
> **Rebuttal by Authors (cont'd)**
>
> ## W2: Early vs. late fusion
> We want to highlight the benefits of our proposed late fusion structure as follows:
>
> 1. Our experimental results already show that, without any fine-tuning of the large models, late fusion generalizes better than early fusion. Across all four LIBERO task suites and the real-robot experiments, we keep both the LLM and the image encoder frozen and rely only on late fusion plus the action decoder, yet we significantly outperform state-of-the-art methods that use early fusion. This directly demonstrates that, under the constraint of no fine-tuning of large models, late fusion provides stronger instruction understanding and generalization.
>
> 1. Early-fusion approaches typically require fine-tuning the pretrained large models, which is at odds with the core motivation of our work. Existing early-fusion architectures rely on deep cross-modal attention that tightly couples vision and language inside the model; consequently, robot policies must heavily fine-tune both the LLM and the vision foundation model, incurring very high training cost and making efficient deployment on edge devices impractical. In contrast, our goal is explicit: to enable already-trained large models to be easily and efficiently deployed on lightweight edge hardware while preserving generalization. We intentionally avoid fine-tuning early-fusion modules, this is precisely the key advantage of NanoVLA.
>
> 1. Our late-fusion module is not merely a simple post-hoc fusion; functionally, it acts as a lightweight multimodal alignment plus a last-layer adaptation mechanism, without modifying the LLM or visual backbone. It serves two key roles in NanoVLA: (1) it aligns vision features and language features in a lightweight way, without requiring the heavy joint fine-tuning and deep coupling used in early fusion; and (2) it performs last-layer task adaptation, our self-attention module adapts the policy to new tasks at low cost, without changing any parameters of the LLM or vision backbone. Thus, for real-world robotic deployment, late fusion offers substantially better deployment properties than early fusion.
>
> 1. We agree that understanding fusion choices matters, but a new VLM-style early-fusion variant is not necessary for this paper. At submission time, compact public VLM options were limited, and constructing a “early-fusion with the same backbones” would create a non-canonical baseline that is not been pretrained by vision language tasks, making it harder to reproduce and less informative than comparing against established early-fusion/VLM systems already in wide use.
>
> We evaluated two widely used early fusion VLA on public dataset: OpenVLA and SmolVLA. Across four LIBERO suites, NanoVLA achieves the best average SR while using a tiny fraction of parameters. Thus, if late fusion significantly “loses” language-vision alignment and damages the generalization, it should underperform early-fusion/VLM counterparts on LIBERO.
>
> To address the concern that our real-world tasks are too simple, we additionally evaluate NanoVLA on a long-horizon manipulation task in a heavily cluttered tabletop environment. The new experiment requires the robot to “first pick up the pink squid and place it in the right bin, then pick up the blue dragon and place it in the left bin”, and must complete two sequential sub-goals while navigating occlusions and distractors. Object and bin poses are randomized per episode. It’s more changing the previous experiments we provided, where the robot needs to accurately identify objects that need to be picked up from all eight different objects in the table top, additionally, the robot needs to know the spatial orientation to put the objects into the correct bin. Additionally, apart from reasoning over multiple sequential sub-tasks, approaching the target, maneuvering around distractors, grasping, and placing, the robot must also complete these tasks while avoiding collisions with clutter.
>
> Figure 4 of the revised manuscript visualizes one successful rollout: the yellow-orange-red trajectory shows the predicted end-effector path, and the frames are sampled at key manipulation events along the episode. This qualitative example demonstrates that NanoVLA can maintain coherent behavior over many steps and operate reliably in visually complex, cluttered scenes, going beyond simple single-object pick-and-place tasks. The updated results are as provided in the Table 3 of the revised manuscript.
>
> For the long-horizon task in complex scenario, NanoVLA maintain high success (76-78%), slightly trailing OpenVLA while remaining far ahead of SmolVLA and $\pi_0$. These results show that our NanoVLA variants not only handle standard single-step manipulations but also generalize well to cluttered, long-horizon, and OOD scenarios, all while operating in a much more parameter-efficient regime.

---

> ### Author Response · Authors · 2025-11-24
> **Rebuttal by Authors (cont'd)**
>
> ## W3: On “LSAC is just receding-horizon control”
> We agree that many works replan periodically; our contribution is a specific plan-long/act-short formulation: the policy trains to predict a long chunk but executes only the first h actions before replanning (h << H). This differs from typical “predict-and-execute the whole chunk” open-loop schemes discussed in our text. As for the receding horizon mechanism proposed in Diffusion Policy, it requires to consume past observations to conduct model predictive control, and then execute few steps of the predicted actions. We provide a simpler solution by only considering the current state to improve the action statbility comparing to action chunking by plan long and act short.
>
> Empirically, our long-short variant shows a flat SR plateau (20-60 steps) with better stability vs. fixed-chunk execution at larger steps, while increasing FPS, which is precisely the edge-deployment sweet spot.
>
> ## W4: Technical novelty
> While each ingredient connects to prior ideas, our edge-oriented integration is new:
> 1. Decoupled late fusion + caching to eliminate redundant per-timestep language compute (Please see our detailed discussion on W2)
> 2. LSAC is proposed to reconcile long-horizon smoothness with short-horizon reactivity. The policy generates a longer chunk of actions, but executes only a short window before re-planning with fresh observations. This amortizes expensive planning over multiple control steps, while keeping behavior smooth and adaptable to new visual evidence.
> 3. MCB router that allocates capacity pre-execution with calibrated win probabilities. The paper positions these under a single principle, spend compute only where it matters for on-device VLA control, and we validates the package on sim + robots.
>
> ## W5: “Will late fusion scale to stronger grounding (e.g., pick any amidst clutter)?”
> We provided two points of evidence in the orignal manuscript to showcase late-fusion's generalization capability:
> 1. LIBERO-Object already stresses object variation: NanoVLA-R achieves 96.2% on this suite (Table 1).
> 2. Real-robot results include precision (open/close lid) and deformable manipulation (banana, towel), where NanoVLA matches or exceeds baselines.
>
> Additionaly, we added a long-horizon manipulation task in a heavily cluttered tabletop environment. Please see our detailed analysis for W2.
>
> W6: Typos and polish
> We thank the reviewer for catching these, we have fix the listed issues (e.g., “OpenVLA-L”, “the the”, Figure 5(a) axis spacing/labels) and do a thorough proofread.
>
> As for “LeRobot was not included the pre-training dataset”, we actually want to express that the pre-training dataset (Bridge Data V2) did not include any rollouts from LeRobot, hence it will be harder for the model to predict the action for a new embodiment.
>
> # Answers to specific questions
> ## Q1. “How many trials are needed to train an accurate router?”
> We following the training dataset to use 50 trails to train the router.
>
>
> ## Q2. “Why does NanoVLA-L underperform compared to NanoVLA-R?”
> NanoVLA-R adapts capacity per task: it defaults to the compact backbone and escalates to the large one only when predicted gains justify the cost. This reduces harmful down-routing while allowing up-routing (which only affects compute, not SR), yielding higher average SR than any single backbone under a limited budget. Concretely, SR rises from 0.805 (L-only) to a broad ≈0.84 plateau with routing, while average model size (compute) drops substantially (Fig. 6).
>
> Intuitively, some short-horizon skills are actually handled better by the smaller LM (lower over-fit to language priors; faster feedback loops), while long-horizon instructions benefit from the larger LM, where routing gets the best of both under a single controller.

---

### Official Review · Reviewer_VDE4 · 2025-11-01

**Soundness:** 3
**Presentation:** 3
**Contribution:** 2
**Rating:** 4
**Confidence:** 3

**Summary:**

This paper introduces NanoVLA, a family of light-weight vision-language-action (VLA) policies  for deployment on edge hardware (e.g., Jetson Orin Nano). The core idea is to rethink inference rather than merely shrink parameters, via three components: (i) late, decoupled fusion of vision and language with caching of instruction features to eliminate redundant cross-modal computation; (ii) long-short action chunking (LSAC), which plans long sequences but executes short sequences; and (iii) dynamic routing that selects a small or large language backbone per task. Together these aim to spend compute only where it matters and keep latency low while preserving generalization.

Empirically, NanoVLA matches or surpasses larger baselines on LIBERO suites and shows strong real-robot results on a LeRobot setup, while achieving dramatic throughput gains on Orin Nano (e.g., 52× higher FPS than OpenVLA under the authors’ setup). The paper includes ablations for LSAC stability and for LLM-feature caching, and formalizes the router with a Beta–Binomial model and Monte-Carlo estimation of pairwise win probabilities.

**Strengths:**

The paper is well-structured with clear motivation. It demonstrates on-edge VLA control with improved success and major latency gains, addressing a potential deployment blocker for household and mobile manipulation. Reported numbers are compelling: SOTA-competitive LIBERO performance with far fewer parameters, strong LeRobot success rates, and notably higher FPS on Orin Nano.

**Weaknesses:**

**LSAC**: While LSAC is effective, the idea of predicting longer sequences and executing shorter sub-segments with periodic replans has been used in Diffusion Policy and Pi0; the paper could better articulate what is novel.

**Routing signal is text-only.** The router is trained as a text-conditioned comparator over models. This assumes that instruction phrasing correlates tightly with task difficulty. However, “pick up the banana” may range from trivial on a clear table to hard in a fruit pile that requires recognition, occlusion handling, and non-prehensile pushes—precisely where a stronger visual backbone is needed. A vision-augmented (or vision+text) router could improve routing fidelity in more realistic scenarios.

**Late fusion vs. early fusion trade-offs.** Decoupling enables caching, but the paper does not directly quantify any loss (or gain) in instruction-following and generalization versus a comparable early-fusion counterpart at equal parameter/compute. A head-to-head ablation—same encoders/action head, with and without early cross-modal attention—reporting instruction sensitivity and OOD generalization would strengthen the claim that late fusion “maintains competitive performance” while improving efficiency.

**Baselines and finetuning.** It is not explicit whether all baselines were finetuned on the same real-robot data mixture and with matched action horizons/chunk sizes. Without this, real-robot gains could partially reflect data or horizon choices rather than architecture.

**Questions:**

First, could you include a direct comparison of late fusion vs. early fusion under matched encoders/params/compute, evaluating instruction following and generalization? This would isolate whether late fusion’s efficiency comes with any semantic trade-offs, beyond the shown caching benefit.

Second, on routing, can you quantify misrouting rates and failure modes, and evaluate a vision-augmented router versus text-only? It would be useful to see thresholds (\tau) vs. success/computation for cluttered vs. clean scenes, and calibration under few-shot task statistics. The Beta–Binomial/MCB formulation is elegant—an ablation versus simpler SR-only routing under distribution shift would be convincing.

Finally, could you clarify baseline training details on real-robot experiments—were all baselines finetuned on the exact same 50-demo/task sets with identical horizons/action parameterization? And could you add a small study on unfreezing the vision encoder to demonstrate whether freezing is always preferable?

---

> ### Author Response · Authors · 2025-11-24
>
> We thank the reviewer for the thoughtful assessment and for highlighting the strengths of our work on edge-side VLAs. Below we clarify points of novelty, provide additional analysis, and experiments.
>
> # W1: On LSAC novelty vs. prior “receding horizon / chunking”
> The key distinction is the horizon-size mismatch between training and inference. By executing only short horizons at inference, the policy remains responsive to new observations, which is an ability that existing action-chunking techniques lack. This differs from prior chunking schemes, which predict a sequence and then execute the entire chunk open-loop, limiting feedback until the next replan. As for the receding horizon mechanism proposed in DP, it consumes past observations to predict actions, and then execute few steps of the predicted actions. We provide a simpler solution by only considering the current state to improve the statbility comparing to action chunking.
>
> # W2: On the router’s “text-only” signal
> Our routing mechanism is designed solely to select the appropriate language model for instruction embeddings; since a lightweight image encoder (we used ResNet-18 for all experiments) already provides sufficiently rich visual features, there is no need for a vision router. The motivation of our dynamic router is to avoid the unnecessary computational overhead of using LLMs when instructions are simple. This design yields robust SR-compute trade-offs: Fig. 6a shows a broad SR plateau (~0.84) for $\tau$∈[0.4,0.9], while Fig. 6b shows substantial compute savings (average model size drops from 520M to ~251M).
>
> # W3: Late fusion vs. early fusion: efficiency and semantics
> Our decoupled design keeps the vision and language encoders frozen, preserving their pretrained semantics, and fuses them once, late with a lightweight transformer. This specific placement of cross-attention (late, not repeated each step) unlocks instruction-feature caching and eliminates redundant per-timestep language compute while still allowing the decoder to bind modalities for action prediction.
>
> We want to highlight the benefits of our proposed late fusion structure as follows:
>
> 1. **Empirical evidence for late fusion.** Our experimental results already show that, without any fine-tuning of the large models, late fusion generalizes better than early fusion. Across all four LIBERO task suites and the real-robot experiments, we keep both the LLM and the image encoder frozen and rely only on late fusion plus the action decoder, yet we significantly outperform state-of-the-art methods that use early fusion. This directly demonstrates that, under the constraint of no fine-tuning of large models, late fusion provides stronger instruction understanding and generalization.
> 1. **Why we avoid early fusion.** Early-fusion approaches typically require fine-tuning the pretrained large models, which is at odds with the core motivation of our work. Existing early-fusion architectures rely on deep cross-modal attention that tightly couples vision and language inside the model; consequently, robot policies must heavily fine-tune both the LLM and the vision foundation model, incurring very high training cost and making efficient deployment on edge devices impractical. In contrast, our goal is explicit: to enable already-trained large models to be easily and efficiently deployed on lightweight edge hardware while preserving generalization.
> 1. **What our late fusion actually does.** Our late-fusion module is not merely a simple post-hoc fusion; functionally, it acts as a lightweight multimodal alignment plus a last-layer adaptation mechanism, without modifying the LLM or visual backbone. It serves two key roles in NanoVLA: (1) it aligns vision features and language features in a lightweight way, without requiring the heavy joint fine-tuning and deep coupling used in early fusion; and (2) it performs last-layer task adaptation, our self-attention module adapts the policy to new tasks at low cost, without changing any parameters of the LLM or vision backbone. Thus, for real-world robotic deployment, late fusion offers substantially better deployment properties than early fusion.
>
> We agree that understanding fusion choices matters, but a new VLM-style early-fusion variant is not necessary for this paper. At submission time, compact public VLM options were limited, and constructing a “early-fusion with the same backbones” would create a non-canonical baseline that is not been pretrained by vision language tasks, making it harder to reproduce and less informative than comparing against established early-fusion/VLM systems already in wide use.
>
> We evaluated two widely used early fusion VLA on public dataset: OpenVLA and SmolVLA. Across four LIBERO suites, NanoVLA achieves the best average SR while using a tiny fraction of parameters. Thus, if late fusion significantly “loses” language-vision alignment and damages the generalization, it should underperform early-fusion/VLM counterparts on LIBERO.

---

> ### Author Response · Authors · 2025-11-24
> **Rebuttal by Authors (cont'd)**
>
> # W4: Baselines, finetuning, and horizons
> All real-robot models (ours and baselines) were finetuned on the same dataset mixture, 50 demonstrations per task across our 10 tasks, using the same action horizon/chunking protocol for fairness. We have made the updates in section 4.2 for clarity.
> Table 3 then reports a apple-to-apple comparison across tasks (including deformable objects and precision lids).
>
> # Answers to your questions
> ## Q1: Late fusion vs. early fusion
> Please see our discussions above
>
> ## Q2 Routing analysis.
> Our routing model is implemented as an LLM that predicts a win probability for each candidate expert given only the natural-language task description. At inference time we always pick the expert with the highest predicted win probability, but the router is intentionally conservative: when it is uncertain or the scores are close, it tends to favor the larger NanoVLA-L expert. In other words, when the router cannot confidently exploit specialization, it gracefully falls back to the strong large model rather than risking a bad choice.
>
> This behavior is also reflected in the OOD setting in Table 3. The “OOD pick-place” task is not seen during router training, yet NanoVLA-R achieves 84.0% success, essentially identical to always using NanoVLA-L (84.0%) and clearly better than the small expert NanoVLA-S (82.0%). The same pattern holds in the average row: NanoVLA-R and NanoVLA-L obtain the same overall performance (85.6%), showing that the router does not degrade accuracy even when some tasks are out-of-distribution. This suggests that, although the win probabilities may be less well calibrated on OOD tasks, they are still accurate enough in a relative sense: whenever the router is unsure how an OOD task relates to the training distribution, it defaults to the safer large expert, so NanoVLA-R matches NanoVLA-L’s performance while still routing many in-distribution tasks to cheaper experts.
>
> For the ablation for MCB versus simpler SR, please refer to Figure 6b, the proposed MCB approach present a wider operating range, while the baseline is highly sensitive to threshold choice. MCB is robust to mis-calibration and task heterogeneity obtain near-maximal precision based on the computational resources, whereas the baseline requires staying very close to $\tau=0$ to avoid a substantial accuracy penalty.
>
> ## Q3: Real-robot baseline training details & unfreezing.
> All real-robot models (ours and baselines) were finetuned on the same dataset mixture, 50 demonstrations per task across our 10 tasks, using the same action horizon/chunking protocol for fairness. We have made the updates in section 4.2 for clarity.

---

### Note · Authors · 2026-01-06

I have read and agree with the venue's withdrawal policy on behalf of myself and my co-authors.